# QUANTUM-INSPIRED TENSORIZED EMBEDDING WITH APPLICATION TO NODE REPRESENTATION LEARNING

## ABSTRACT

Node representation learning a.k.a. network embedding (NE) is an essential technique for network analysis by representing nodes as vectors. Most existing NE algorithms require a $O(p)$ space complexity for each node embedding ($p$ is the final embedding vector size). Impressed and inspired by the large Hilbert space of quantum systems, we propose a brand new NE algorithm *node2ket* and its variant *node2kek+* by imitating behaviors of quantum systems. Theoretically, we give analysis on how it unifies existing embedding methods including both conventional ones and tensorized ones, and prove that our methods achieve to use $O(p^{1/\ln p} \ln p)$ or even smaller space for each node embedding. We are the first to successfully conduct node embedding in the super large Hilbert space. Experiments are conducted on five public real-world networks where methods are evaluated through the tasks of network reconstruction, link prediction, and node classification. On BlogCatalog, our method achieves to outperform all baselines with 1/32 training parameters and 1/16 running time on the same machine. On DBLP, the reconstruction precision of node2ket achieves to be 3 times higher than the best baseline i.e. LouvainNE. The source code will be publicly available.

*Hilbert space is a big place!*

– Carlton Morris Caves.

## 1 INTRODUCTION

Node representation learning, also known as network embedding (NE), has been well-studied in decades. Borrowing idea from word embedding in NLP, deepwalk (Perozzi et al., 2014) first proposes to learn node embeddings for downstream tasks by feeding random walk sequences to word2vec (Mikolov et al., 2013). Afterward, numerous NE methods for different types of networks (e.g. heterogeneous/multiplex networks) and for different purposes (e.g. preserving structural similarity) are proposed (see Sec. 2 for more). Despite the great success, the existing NE methods still suffer from the high space complexity for storing/training the embeddings whose size grows linearly to the multiplication of the number of nodes and the embedding dimension (Xiong et al., 2022) (see Appendix A for details). As a result, there is difficulty in deploying these methods on memory-limited devices for privacy-sensitive computing e.g. recommender systems in social networks on mobile phones. Meanwhile, the expressiveness of the embedding vectors can be restricted.

Recently, inspired by quantum mechanics, word2ket (Panahi et al., 2020) was proposed for compressive word embedding by imitating the behaviors of quantum bits in quantum systems. By tensor entanglement, word2ket achieves both a high compressive ratio and experimental results comparable to conventional embeddings. Though providing many insights in generating embeddings, word2ket still faces difficulties transferring to embedding problems in other fields such as node representation learning (Hamilton et al., 2017b). These difficulties include: **i)** Word2ket generates embeddings from a fixed complete binary tree, which **neglects the latent structure of the input data and hence brings challenges to learning embeddings for data with explicit structures (e.g. networks). ii)** Word2ket adopts deep neural networks to decode the embedding and is trained in a supervised way, **which is rather different from many embedding methods** (Tang et al., 2015; Grover & Leskovec, 2016) that decode embeddings via simply inner product operation and train embeddings by optimizing the objective based on Noise Contrastive Estimation (NCE) (Gutmann & Hyvärinen, 2010) in a self-supervised (contrastive learning) style.

In this paper, we propose a novel network embedding paradigm with special treatments concerning the above two problems. **For the embedding generation,** we propose an embedding module named as *Tensorized Embedding Blocks* (TEBs) composed of columns of *Tensor Units* (TUs), each representing an embedding tensor. TEBs can unify the embedding layers in word2ket/word2ketXS by making changes in the way of indexing TUs (see Sec. 4.1 and Sec. 4.5). A good strategy of assigning TU indices to the embedded nodes will result in a model with both high expressiveness and space efficiency, which is proved by the experiments. **For the embedding learning,** we optimize the widely adopted NCE-based objective and succeed in learning node embeddings in a $2^{32}$-dimensional Hilbert space stably and robustly on a single machine (details in Sec. 4.2). Specifically, we propose two variants on different compressive levels, named as *node2ket* and *node2ket+*. The former is a preliminary baseline and the latter is more compressive by utilizing graph partition techniques. Furthermore, we give a complexity analysis of the embedding modules in TEBs, and theoretically prove an upper bound of the minimum of the required space to store/train embeddings for node2ket and node2ket+ in Sec. 4.3 and Sec. 4.4 respectively. **Our contributions are listed as follows:**

1) We are the first to formalize a general embedding architecture, Tensorized Embedding Blocks (TEBs), which generate embeddings in Hilbert space by tensor product and entanglement, imitating behaviors of quantum systems. In this way, high-dimensional embeddings of high expressiveness in the Hilbert space can be obtained via a small number of parameters. Our tensorized embedding architecture is a generalization of the recent method word2ket/word2ketXS and also unifies conventional embedding methods.

2) We take an initiative (the first time to our best knowledge) to adopt tensorized node embedding in representation learning based on Noise Contrastive Estimation. Specifically, we propose two variants at different compressive levels, *node2ket* and *node2ket+*, with the bound of compressive power proved respectively. Compared with the preliminary baseline node2ket, node2ket+ further utilizes graph partition techniques in the design of TEBs, in which way features on different levels of granularity are preserved and the embedding model becomes more compressive.

3) Experiments on network reconstruction and link prediction show the overwhelming performance of node2ket. Notably, in the network reconstruction experiments, it becomes the top method on BlogCatalog with 1/32 training parameters and 1/16 running time, and the precision on DBLP is more than 3 times higher than the best one of baselines i.e. LouvainNE (Bhowmick et al., 2020).

## 2 RELATED WORKS

**Network Embedding.** Here we mainly discuss the NE methods based on the language model word2vec (Mikolov et al., 2013), which are often designed for different types of networks, e.g. homogeneous networks (Perozzi et al., 2014; Grover & Leskovec, 2016; Tang et al., 2015), heterogeneous networks (Dong et al., 2017) and multiplex networks (Liu et al., 2017; Qu et al., 2017; Xiong et al., 2021). It also often aims to preserve specific types of information, e.g. low-order proximity (Tang et al., 2015), structure similarity (Ribeiro et al., 2017), versatile similarity measures (Tsitsulin et al., 2018), and cross-network alignment relations (Du et al., 2022). Apart from NE methods derived from word2vec, methods based on deep neural networks (Wang et al., 2016) especially graph neural networks (Hamilton et al., 2017a) are also influential in literature, which, however, are technically divergent to this paper.

**Tensorized Embedding Model.** Tensors are a generalization of vectors and matrices. Thanks to the capability of representing and manipulating high-dimensional subjects, tensor has been successfully applied to exploit complex physical systems (Nielsen & Chuang, 2002). The most crucial application is to understand the quantum many-body systems and to make an efficient description of quantum states residing in the exponential growth of the Hilbert space (Bridgeman & Chubb, 2017). In addition, tensor networks composed of multiple tensors can also be used to understand many of the foundational results in quantum information (Orús, 2019). Utilizing the tensor network structure makes it reasonably possible to simplify the concept like quantum teleportation, purification, and the church of the larger Hilbert space (Bridgeman & Chubb, 2017). Inspired by this, many tensorized embedding models try to take advantage of tensor operations like contraction and decomposition to handle large volumes of data and compress parameters, ranging from image classification (Stoudenmire & Schwab, 2016; Martyn et al., 2020; Selvan & Dam, 2020; Cheng et al., 2021) to natural language processing (Panahi et al., 2020; Ma et al., 2019; Liu et al., 2020; Qiu & Huang, 2015). It

is anticipated that the tensor concept introduced by quantum physics will give us new perspectives on designing the structure of deep learning models and comprehending the learning capability of artificial neural networks (Levine et al., 2017).

## 3   PRELIMINARIES AND BACKGROUND

### 3.1   NETWORK EMBEDDING BASED ON NOISE CONTRASTIVE ESTIMATION

Given a network described by the node set $\mathcal{V}$ and the edge set $\mathcal{E}$, node representation learning or network embedding aims to represent each node $i \in \mathcal{V}$ as a fixed-length vector $\mathbf{x}_i \in \mathbb{R}^p$. To fully explore network structures for different purposes, a *corpus* $\mathcal{D}$ is usually created as the input data of embedding training. Typically, $\mathcal{D}$ can be the edge set (Tang et al., 2015), random walk sequences (Perozzi et al., 2014; Ribeiro et al., 2017; Du et al., 2022), etc.

**Objective Function.** Initially, the objective is to maximize the following log softmax function:

$$\sum_{(i,j)\in\mathcal{D}} \log \frac{\exp(\mathbf{x}_i^\top \mathbf{x}_j)}{\sum_{j'\in\mathcal{V}} \exp(\mathbf{x}_i^\top \mathbf{x}_{j'})}, \tag{1}$$

where the positive pairs $(i, j)$ are sampled from $\mathcal{D}$. However, Eq. 1 is computationally intractable for a large $\mathcal{V}$, and thus surrogate objectives are proposed, including Hierarchical Softmax and Negative Sampling (Mikolov et al., 2013). In this paper, we only discuss a typical objective based on the widely adopted Negative Sampling:

$$\sum_{(i,j)\in\mathcal{D}} \Big[ \underbrace{\log \sigma(\mathbf{x}_i^\top \mathbf{x}_j)}_{\text{positive samples}} + \underbrace{\sum_{k=1}^{K} \mathbb{E}_{j'\sim\mathbb{P}_n} \log \sigma(-\mathbf{x}_i^\top \mathbf{x}_{j'})}_{K \text{ negative samples for each positive sample}} \Big], \tag{2}$$

where $\sigma(x) = 1/(1 + \exp(-x))$ is sigmoid function, $K$ is the number of negative samples for each positive sampled pair, $\mathbb{P}_n$ is the distribution of negative samples $j'$ which is usually set proportional to node degrees.

### 3.2   QUANTUM MECHANICS AND TENSORIZED EMBEDDING

**Quantum Mechanics.**   Quantum states are usually written in a 'bra-ket' notation. In matrix representations, **kets** are column vectors denoted as $|\psi\rangle$ (*where the name word/nodeket comes*) and **bras** are the conjugate transpose of the kets denoted as $\langle\psi|$. The state of a single quantum bit (qubit) $|\psi\rangle$ is defined in a 2-dimensional Hilbert space $\mathbb{C}^2$, described by a superposition of two orthonormal basis states $|0\rangle = (1,0)^\top$ and $|1\rangle = (0,1)^\top$, $|\psi\rangle = \alpha|0\rangle + \beta|1\rangle$, where $\alpha, \beta \in \mathbb{C}$ are probability amplitudes satisfying $|\alpha|^2 + |\beta|^2 = 1$. The state of an $N$-qubit quantum system is in the Hilbert space $\otimes_{i=1}^N \mathbb{C}^2$, where $\otimes$ denotes tensor product. Considering two quantum systems $A$ and $B$ with corresponding Hilbert spaces $\mathcal{H}_A$ and $\mathcal{H}_B$, the composite space of $A$ and $B$ is $\mathcal{H} = \mathcal{H}_A \otimes \mathcal{H}_B$. If we use $|\phi\rangle_A$ and $|\phi\rangle_B$ to denote states of system $A$ and $B$ respectively, then a state $|\phi\rangle \in \mathcal{H}$ that can be written as $|\phi\rangle = |\phi\rangle_A \otimes |\phi\rangle_B$ is called a **pure state**, otherwise it is an **entangled state**.

**Tensorized Embedding by Word2ket.**   Word2ket (Panahi et al., 2020) conducts word embedding by imitating behaviors of quantum systems. Embedding of each word is represented as an entangled quantum state in the Hilbert space, which is the summation of several pure state embeddings. A pure state embedding is formed by tensor product between some lower-dimensional tensors. During the embedding procedure, some simplifications are adopted: i) real vectors are used as embeddings instead of complex ones in the quantum states; ii) the tensor product between tensors is implemented as the Kronecker product between vectors (it does not affect the results mathematically, since a vector can be deemed as a special tensor (Bridgeman & Chubb, 2016), and Kronecker product can be viewed as a form of vectorization of the tensor product); and iii) when imitating quantum entanglements, it uses learned tensors but not orthonormal basis states for implementation convenience, which means that a generated entangled state embedding may not be a strictly entangled one. These simplifications are also adopted by this paper.

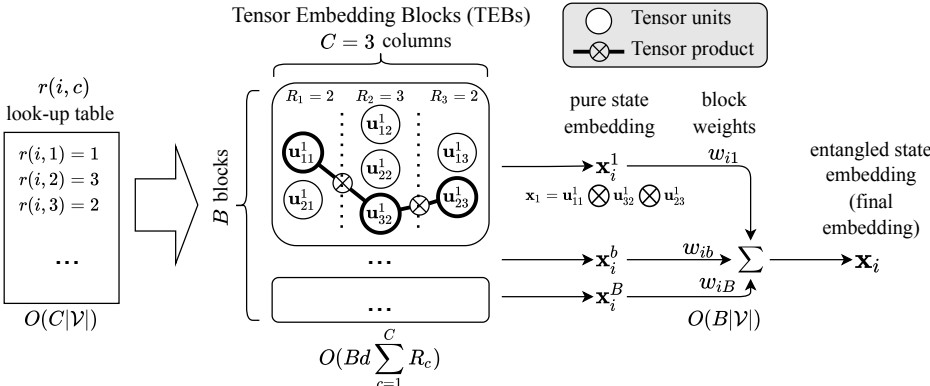

Figure 1: A toy example to generate embedding from the given tensorized embedding blocks. To get the embedding $\mathbf{x}_i$, the first step is to look $r(i,c)$ up from the table at the left-hand side, and then pick the tensor units out from the TEBs and calculate the final embedding according to Eq. 3.

## 4 TENSORIZED NETWORK EMBEDDING

Sec. 4.1 first introduces how we obtain embeddings from the proposed architecture Tensorized Embedding Blocks (TEBs). In Sec. 4.2 we introduce how to train embeddings in TEBs efficiently. Then in Sec. 4.3 and Sec. 4.4 we introduce the specific network embedding methods node2ket and node2ket+ respectively. Finally in Sec. 4.5 we discuss the relation of the framework to other embedding methods.

### 4.1 EMBEDDING VIA TENSORIZED EMBEDDING BLOCKS

**Tensorized Embedding Blocks (TEBs).** Before introducing how node2ket works, we first propose a novel embedding module named as Tensorized Embedding Blocks. A TEB is composed of a certain number of Tensor Units (TUs), and each TU represents an embedding vector in $\mathbb{R}^d$. A TEB has $C$ columns where the $c$-th column has $R_c$ TUs. We denote the embedding vector of the TU located at the $r$-th row and the $c$-th column of the $b$-th block by $\mathbf{u}^b_{rc}$.

**Embeddings Generation.** We use $B$ TEBs for embedding generation. Though more freedom can be added to the number of rows $\{R_c\}_{c=1}^C$ for different TEBs, we assume that they are the same without losing generality. For node $i$, we obtain the embedding $\mathbf{x}_i$ by:

$$\mathbf{x}_i = \sum_{b=1}^B w_{ib}\mathbf{x}_i^b, \quad \mathbf{x}_i^b = \bigotimes_{c=1}^C \mathbf{u}^b_{r(i,c)c}, \tag{3}$$

where $\mathbf{x}_i^b$ is the pure state embedding in a $\mathbb{R}^{d^C}$-dimensional Hilbert space given by the $b$-th TEB for node $i$, and $w_{ib}$ is the corresponding block weight of node $i$ on the $b$-th TEB. The $r(i,c)$ in the subscription of TU embeddings $\mathbf{u}^b_{r(i,c)c}$ indicates that the row index $r$ depends on the node $i$ and the column index $c$. For better understanding, we give a toy example of TEBs and how to generate $\mathbf{x}_i$ according to the given $r(i,c)$ look-up table as shown in Fig. 1.

**Complexity Analysis.** There are three parts of parameters: i) the $r(i,c)$ table has $O(C|\mathcal{V}|)$ parameters, ii) the embeddings of TUs have $O(Bd\sum_{c=1}^C R_c)$ parameters, and iii) the block weights have $O(B|\mathcal{V}|)$ parameters. The used parameters is totally $O(Bd\sum_{c=1}^C R_c + C|\mathcal{V}| + B|\mathcal{V}|)$. For more detailed complexity analysis please refer to Sec. 4.3 and Sec. 4.4.

### 4.2 MODEL LEARNING

In this section, we show how to efficiently train the generated tensorized embedding by TEBs via updating embeddings directly according to computed gradients, without automatically forward/backward propagation as most DL models did. In this way, we save the computation time/space from $O(d^C)$ to $O(dC)$, firstly making learning embeddings in the high-dimensional Hilbert space feasible. Details are put in Appendix C.2, including gradient derivation and the complete algorithm.

**Gradient.** During the model optimization, we do not need to calculate the full node embeddings. Instead, we compute the gradient and update TU embeddings directly without a complete procedure of forward propagation. Given two embeddings $\mathbf{x}_i$ and $\mathbf{x}_j$:

$$\mathbf{x}_i = \sum_{b=1}^{B} w_{ib} \bigotimes_{c=1}^{C} \mathbf{u}_{r(i,c)c}^{b}, \quad \mathbf{x}_j = \sum_{b=1}^{B} w_{jb} \bigotimes_{c=1}^{C} \mathbf{u}_{r(j,c)c}^{b}, \tag{4}$$

and the label $l$ ($l = 1$ for positive samples and $l = 0$ otherwise), we have the gradients for TU embeddings according to the objective Eq. 2:

$$\frac{\partial \log \sigma(2(\mathbb{1}_l - 0.5)\langle \mathbf{x}_i, \mathbf{x}_j \rangle)}{\partial \mathbf{u}_{r(i,c)c}^{b}} = w_{ib}\big[\mathbb{1}_l - \sigma(\langle \mathbf{x}_i, \mathbf{x}_j \rangle)\big] \sum_{b'=1}^{B} w_{jb'} \mathbf{u}_{r(j,c)c}^{b'} \prod_{c' \neq c} \langle \mathbf{u}_{r(i,c')c'}^{b}, \mathbf{u}_{r(j,c')c'}^{b'} \rangle, \tag{5}$$

where $\mathbb{1}_l$ is an indicator returning 1 if $l = 1$ and 0 otherwise. The gradients for block weights are:

$$\frac{\partial \log \sigma(2(\mathbb{1}_l - 0.5)\langle \mathbf{x}_i, \mathbf{x}_j \rangle)}{\partial w_{ib}} = \big[\mathbb{1}_l - \sigma(\langle \mathbf{x}_i, \mathbf{x}_j \rangle)\big] \sum_{b'=1}^{B} w_{jb'} \prod_{c=1}^{C} \langle \mathbf{u}_{r(i,c)c}^{b}, \mathbf{u}_{r(j,c)c}^{b'} \rangle. \tag{6}$$

For detailed derivation of the gradients, please see Appendix C.1.

**Model Training.** We optimize the objective via Asynchronous Stochastic Gradient Descent (Recht et al., 2011) on CPUs as most works in NCE-based embedding (Mikolov et al., 2013; Tang et al., 2015) and the famous open-source library gensim[1] do. Technical details are given in Appendix C.2.

### 4.3 NODE2KET: A PRELIMINARY BASELINE

For node2ket (namely n2k), we set $r(i,c) = i$ as constants, and $R_1 = R_2 = \cdots = R_C = |\mathcal{V}|$. It means we don't need space to store the $r(i,c)$ table. We give a constraint over the embedding dimension $d$ in Assumption 1, according to which we can reach Theorem 1 where the ultimate compressive power is analyzed. The proof and further comments are given in Appendix D.

**Assumption 1.** [**Dimensional Assumption**] *If a $p$-dimensional final embedding is required, the dimension of generated embedding should not be smaller than $p$, i.e. $d^C \geq p$;*

**Theorem 1.** [**Compressive Power of Node2ket**] *Given node set $\mathcal{V}$ and the final embedding dimension $p$, when $r(i,c) = i$ are constants (indicating $R_c = |\mathcal{V}|, \forall 1 \leq c \leq C$) and use block weights as trainable parameters, the total number of parameters is #params $= B(C|\mathcal{V}|d + |\mathcal{V}|)$. Omitting the integer constraints of $d$ and $C$, we obtain the minimum number of used parameters of the whole embedding model as:*

$$\min \#params = B|\mathcal{V}|p^{1/\ln p} \ln p + B|\mathcal{V}|.$$

*When the block weights are set as constants, #params $= BC|\mathcal{V}|d$, and we have:*

$$\min \#params = B|\mathcal{V}|p^{1/\ln p} \ln p.$$

### 4.4 NODE2KET+: COMPRESSIVE EMBEDDING UTILIZING GRAPH PARTITION

To further compress node2ket, we propose node2ket+ (n2k+) utilizing graph partition techniques. Specifically, we set $r(i,c) = \text{Par}_c(i)$ where $\text{Par}_c(\cdot)$ is the $c$-th partition algorithm and $\text{Par}_c(i)$ returns the index of the partition that node $i$ belongs to. In this way, $R_c$ is equal to the number of partitions given by the function $\text{Par}_c$. In the implementation, we use Louvain partition (Blondel et al., 2008) with different settings of resolution as the graph partition algorithm, which is a scalable algorithm aiming to maximize the modularity of the partitions. We control that $R_1 \leq R_2 \leq \cdots R_C$, hence the model can extract structure information on different levels of granularity. Theoretically, we further give Assumption 2 over the size of TEBs, and then give a bound of the ultimate compressive power of node2ket+ in Theorem 2 based on Assumption 1 and 2. The proof for the following Theorem 2 is presented in Appendix D by borrowing the tool of asymptotic analysis (Bender & Orszag, 2013), which is technically nontrivial and new to the network embedding community.

**Assumption 2.** [**Combinatorial Assumption**] *A TEB should be capable to generate at least $|\mathcal{V}|$ different pure state embeddings, which indicates $\prod_{c=1}^{C} R_c \geq |\mathcal{V}|$.*

---

[1] https://radimrehurek.com/gensim/index.html

Table 1: Differences between word2ket(XS) and node2ket(+) (no downstream tasks are involved).

| | Step 2: Embedding Generation | Step 3: Embedding Learning (Distributed Updating) | | | | Step 1: Prepare Input Data |
|---|---|---|---|---|---|---|
| | | Decoder | Objective | Scalable | Emb. Dim. $p$ | |
| word2ket word2ketXS | special case of node2ket | neural networks | supervised | ✗ | 8000* | input sentences (unstructured data) |
| node2ket node2ket+ | flexibly defined | inner product | noise contrastive estimation | ✔ | $2^{32}$ | input links (structured data) |
| The dimension 8000* is reported in the official paper. | | | | | | |

**Theorem 2.** [**Compressive Power of Node2ket+**] *Given the node set $\mathcal{V}$ and the final embedding dimension $p$, when node2ket utilizes both the $r(i,c)$ table and block weights, the total number of parameters is #params $= Bd\sum_{c=1}^{C} R_c + B|\mathcal{V}| + C|\mathcal{V}|$. We obtain the minimum number of used parameters of the whole embedding model:*

$$\min \#params < BC^*\big(p|\mathcal{V}|\big)^{1/C^*} + B|\mathcal{V}| + C^*|\mathcal{V}|,$$

*where $C^* = \sqrt{\Big[(\ln p|\mathcal{V}|)^2 - \frac{1}{2}\Big(\ln p|\mathcal{V}| + \frac{1}{p|\mathcal{V}|}\Big)\Big]/(|\mathcal{V}|/B + 1)}$. When the block weights are set as constants, we have:*

$$\min \#params < BC^*\big(p|\mathcal{V}|\big)^{1/C^*} + C^*|\mathcal{V}|. \tag{7}$$

### 4.5 RELATION TO OTHER EMBEDDING METHODS

**Relation to Conventional Embedding.** When $B = 1$, $C = 1$, the embeddings of generated by node2ket become $\mathbf{x}_i = \mathbf{u}_{i1}^1$ where no tensor product or entanglement is utilized and the embedding degrades to the conventional embedding. The embedding look-up step in traditional NE methods becomes a special case in the corresponding step of node2ket though the training is different.

**Advantages over Word2ket.** A complete network embedding procedure basically has three steps: i) prepare the input data, ii) generate embedding for the input data, and iii) learn the embeddings. We analyze the advantages of node2ket over word2ket from the three aspects (details in Appendix B.1 and see summarization in Table 1).:

**i) Node2ket can embed various explicitly structured data beyond sequence.** It is achieved by defining $r(i,c)$ to fit the structures of input data utilizing graph partition techniques to encode structures of data into TEBs (e.g. by setting $r(i,c) = r(j,c)$ if node $i$ and $j$ belong to the same community in node2ket+), while word2ket can not utilize inherent structures of input data.

**ii) Node2ket has better space efficiency due to the flexible definition.** Embedding generation modules of word2ket/word2ketXS are special cases of our proposed TEBs by setting $w_{ib} = $ constant and $C = 2^n$. The simplification of word2ket reduces the representational ability of the embeddings and also limits the space efficiency of word2ket.

**iii) Scalable learning by NCE without explicitly maintaining embedding $\mathbf{x}_i$.** Node2ket is distributed trained based on the NCE objective Eq. 2. During training, by iteratively updating the embedding $\mathbf{u}_{r(i,c)c}^b$, the process avoids computing the embedding $\mathbf{x}_i$ and achieves to conduct embedding in the super high-dimensional Hilbert space. In comparison, Word2ket/word2ketXS train embeddings in a supervised way with an NN decoder, which requires computing the full embedding of $\mathbf{x}_i$ and hence fails to do embedding in high-dimensional space. Experimentally node2ket achieves an embedding dimension of $2^{32}$ (can be indeed more than $2^{128}$ on our machine) while word2ket only reports an 8000 embedding dimension.

## 5 EXPERIMENTS

In experiments, we investigate downstream tasks including network reconstruction (Sec. 5.2), link prediction (Sec. 5.3), and node classification (Appendix E.6). In Sec. 5.4, we conduct ablation studies to investigate the roles of different parts in the model. In Sec.5.5, good parallelizability of the methods is shown. And in Appendix E.5, we further display the overwhelming performance of our methods on network reconstruction by visualization of the reconstructed results.

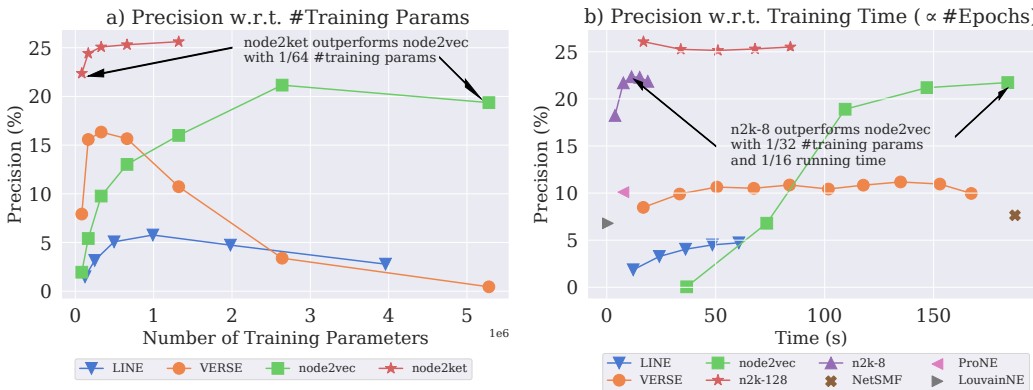

Figure 2: Comparison on BC. n2k-8/128 denotes n2k by $C$=2, $d$=4 and $C$=8, $d$=16 respectively.

## 5.1 EXPERIMENTAL SETTINGS

All of the experiments are run on a single machine with Intel(R) Core(TM) i9-7920X CPU @ 2.90GHz with 24 logical cores and 128GB memory.

**Datasets.** We investigate five publicly available real-world datasets of different scales, different densities, and containing different types of information. All of the networks are processed as undirected ones. The detailed statistics of the networks are given in Table 7 in Appendix E.2. We give a brief introduction of the datasets as follows: **YouTube** (Tang & Liu, 2009b) is a social network with millions of users on youtube.com; **BlogCatalog** (Tang & Liu, 2009a) (BC) is a social network of bloggers who have social connections with each other; **PPI** (Breitkreutz et al., 2007) is a subgraph of the PPI network for Homo Sapiens; **Arxiv GR-QC** (Leskovec et al., 2007) is a collaboration network from arXiv and covers scientific collaborations between authors with papers submitted to General Relativity and Quantum Cosmology category; **DBLP** (Ley, 2002) is a citation network of DBLP, where each node denotes a publication, and each edge represents a citation.

**Baselines.** The investigated NE methods include three based on NCE, two based on matrix factorization, and one based on community detection. Here we introduce them briefly: **LINE** (Tang et al., 2015) models first-order proximity (connection strengths) and second-order proximity (similarity of neighborhoods) of given networks, and technically feed the adjacency matrix to two types of NCE-based objectives to learn node embeddings; **node2vec** (Grover & Leskovec, 2016) designs a biased random walk algorithm to simultaneously preserve Bread-First Search information and Depth-First Search information, and feed the random walk sequences to word2vec; **VERSE** (Tsitsulin et al., 2018) feeds node similarities (Personalized PageRank, Adjcency Similarity, and SimRank) to the NCE-based objective; **NetSMF** (Qiu et al., 2019) obtains node embeddings by sparse matrix factorization on a sparsified dense random walk matrix; **ProNE** (Zhang et al., 2019) initializes node embeddings by sparse matrix factorization, and then enhances the embeddings via spectral propagation; **LouvainNE** (Bhowmick et al., 2020) first adopts Louvain graph partition algorithm (Blondel et al., 2008) and then learn node embeddings from the partitions. HHNE (Zhang et al., 2021) trains hyperbolic node embeddings for heterogenous networks because hyperbolic geometry can reflect some properties of complex networks.

**Parameter Settings.** For all methods that can set the number of threads (node2ket and baselines except ProNE and LouvainNE), we use 8 threads as the default. Embedding dimension is set to 128 as the default. For node2ket, we set the number of blocks $B = 1$, number of negative samples $K = 1$ by default. For more details of parameter settings, please refer to Appendix E.3.

## 5.2 EXPERIMENTS ON NETWORK RECONSTRUCTION

Network reconstruction is used to measure an embedding model's ability to preserve nodes' origin local structure. In line with (Wang et al., 2016), for baseline methods, we measure the score of a pair of node embeddings by negative Euclidean distance. While computing the Euclidean distance between two embeddings in the $d^C$-dimensional Hilbert space is computationally intractable, we use the inner product instead. During the reconstruction process, we calculate the scores between

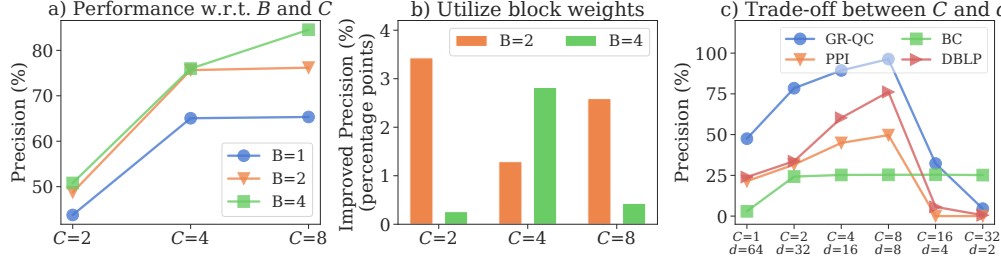

Figure 3: Ablation studies on the number of TEBs $B$, column number $C$, TU embedding dimension $d$, and the adoption of block weights with different $B$ and $C$. Tests of a) and b) run on GR-QC.

Table 2: Network reconstruction precision on medium-scale networks. The value in brackets (e.g. 0.70 with superscript $*$ at the first row) is the ratio of used parameters by node2ket+ to store embeddings against traditional NE methods: $d \sum_{c=1}^{C} R_c / pN$.

| dataset | LINE | node2vec | VERSE | NetSMF | ProNE | LouvainNE | HHNE | n2k | n2k+ |
|---|---|---|---|---|---|---|---|---|---|
| GR-QC | 54.19 | 66.38 | 56.08 | 30.26 | 56.37 | 43.47 | 50.50 | **97.31** | 92.43 (0.70*) |
| BC | 4.80 | 21.16 | 10.96 | 7.64 | 10.10 | 6.79 | 0.69 | **26.07** | 25.06 (0.90) |
| DBLP | 2.34 | 21.17 | 16.96 | 18.08 | 17.76 | 21.33 | 11.40 | 86.40 | **86.62** (0.77) |
| PPI | 15.18 | 48.13 | 27.59 | 17.14 | 24.02 | 17.77 | 18.23 | 74.07 | **82.25** (1.00) |

all possible pairs of the embeddings, namely $|\mathcal{V}|(|\mathcal{V}| - 1)/2$ undirected node pairs. Then we sort the scores and select the top-$|\mathcal{E}|$ node pairs, denoted as $\mathcal{E}'$. Then the precision is defined as Precision = $|\mathcal{E} \cap \mathcal{E}'|/|\mathcal{E}|$. For large-scale networks, we use Prec@N to approximately measure the precision (for detailed explanation please see Appendix E.1).

**Evaluation on Medium-scale Networks:** We set the embedding dimension as 128 for all the baselines. For node2ket, we set $B = 1$, $C = 8$, and $d = 16$, which means that to store the embeddings, the number of used parameters for each node is $BCd = 128$, the same as the baselines. For node2ket+, we also set $B = 1$, $C = 8$, and $d = 16$. Note that node2ket+ decreases the number of TUs but brings space consumption of the $r(i, c)$ table, so the average consumed space to store embedding for each node may not be equal to 128. The experimental results are given in Table 2, which shows that **i) both node2ket and node2ket+ achieve the precision that conventional embedding methods can hardly reach by efforts in feature engineering as most works do;** ii) even after decreasing the number of TUs, node2ket+'s performance does not decrease significantly compared with node2ket, indicating that further compression without sacrificing performance is still possible.

**Evaluation on Large-scale Network (YouTube):** We only evaluate baselines that are scalable and could be run on our machine. We set the embedding dimension as 4 for baselines. Since node2ket fails to generate embeddings because it requires TU dimensions $d \geq 4$ and $C \geq 2$, we only evaluate node2ket+, whose used space to store embeddings is a little less than baselines (compressive ratio=0.99). Results are given in Table 3, showing that node2ket+ outperforms baselines mostly and also converges faster compared with other NCE-based embedding methods.

**Efficiency Evaluation and Scalability.** We conduct the experiments on BlogCatalog. Two questions are raised: to outperform baselines, i) how many training parameters does node2ket need? and ii) how much time does node2ket take? Answers are in Fig. 2 – node2ket achieves to outperforms the best baselines **with 1/64 training parameters** in the evaluation of required training parameters, and with **1/32 training parameters and 1/16 running time** in the evaluation of running time. The results show both the efficiency on space saving and time saving. The **scalability** has been proved in the experiments on YouTube (Table 3), where it takes about two minutes for the running of node2ket+ on a network of 1 million nodes with 8 CPU threads.

### 5.3 EXPERIMENTS ON LINK PREDICTION

We randomly mask 1% edges of the given network, and create the same amount of edges as noise edges by randomly connecting nodes in the network. Then, we conduct network embedding on the network where masked edges are removed, and predict the masked edges from the mixture of masked edges and noise edges. The used metric is the same as network reconstruction (Sec. 5.2). Experimental results in Table 4 show that node2ket outperforms all the baselines consistently.

Table 3: Network reconstruction precision on the large-scale network, YouTube.

| | Method | Prec@2 | Prec@10 | Prec@50 | Running Time (s) |
|---|---|---|---|---|---|
| NCE-based | LINE (Tang et al., 2015) | 71.35 | 36.54 | 16.92 | 502.67 |
| | node2vec (Grover & Leskovec, 2016) | 44.85 | 9.80 | 3.14 | 3979.52 |
| | VERSE (Tsitsulin et al., 2018) | 53.68 | 34.20 | 25.08 | 435.03 |
| Others | ProNE (Zhang et al., 2019) | 76.80 | 41.79 | 18.96 | 85.71 |
| | LouvainNE (Bhowmick et al., 2020) | 85.74 | 70.89 | **63.12** | 8.63 |
| | node2ket+ (ours) | **87.51** | **74.75** | 61.69 | 122.81 |

Table 4: Link prediction precision comparison on medium-scale networks.

| dataset | LINE | node2vec | VERSE | NetSMF | ProNE | LouvainNE | HHNE | node2ket |
|---|---|---|---|---|---|---|---|---|
| GR-QC | 95.86 | 91.72 | 93.79 | 87.59 | 93.79 | 86.90 | 93.10 | **97.24** |
| BC | 53.08 | 77.58 | 33.35 | 59.34 | 75.27 | 57.99 | 38.77 | **81.95** |
| DBLP | 75.45 | 82.09 | 68.81 | 82.09 | 85.31 | 73.44 | 85.92 | **91.95** |
| PPI | 55.15 | 77.58 | 51.55 | 72.16 | 73.97 | 57.99 | 65.72 | **79.12** |

## 5.4 ABLATION STUDY

We conduct ablation studies on node2ket, while the conclusions are the same for node2ket+.

**Number of TEBs $B$ and Number of Columns $C$.** Experiments of network reconstruction are conducted on GR-QC. We set $w_{ib} = 1/\sqrt{B}$ as constants, $d = 4$, and run node2ket with $C = 2, 4, 8$ and $B = 1, 2, 4$. Results are shown in Fig. 3 a), which shows that model performance improved when $C$ or $B$ increases when $d$ is small. However, we also observe that when $d$ is larger, increasing $B$ will have a negative influence on model performance.

**Use Block Weights $w_{ib}$ as Trainable Parameters.** Experiments of network reconstruction are conducted on GR-QC. We set $d = 4$, and run node2ket with $C = 2, 4, 8$ and $B = 2, 4$. In Fig. 3 b) we show the improved precision compared with setting $w_{ib}$ as constants, which proves the positive roles of $w_{ib}$ in improving model's expressiveness when $d$ is small. However, when $d$ is larger, trainable $w_{ib}$ is likely to make a negative influence on model performance.

The experimental results for the above two studies when $d = 8$ are in Table 8 in Appendix E.4.

**Trade-off between Number of Columns $C$ and Embedding Dimension $d$.** Experiments of network reconstruction are conducted on all the medium-scale datasets. We fix $Cd = 64$ and run node2ket with $C = 1, 2, 4, 8, 16, 32$, and plot the results in Fig. 3 c). We see that **i) when $C = 1$, i.e. the tensorized embedding model is degraded to the conventional embedding model, the model performance decreases,** and ii) when $d \leq 4$, the performance is also negatively influenced.

## 5.5 PARALLELIZABILITY

We evaluate node2ket on DBLP. We set $C = 8$, $d = 8$, and run the same number of SGD iterations with the thread number varying from 1 to 16. We record the running time and precision as shown in Fig. 4. The results show that i) the model performance is stable under the setting of different numbers of threads, and ii) the running time $t$ almost decrease linearly w.r.t. the number of threads when the number of threads is not larger than 16.

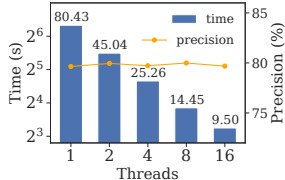

Figure 4: Parallelizability.

## 6 CONCLUSIONS AND FUTURE WORK

We have proposed the network embedding methods *node2ket* and *node2ket+*, by training the embedding module *Tensorized Embedding Blocks* with an objective based on Noise Contrastive Estimation.

**Limitation & future works.** With the (possible) arrival of Noisy Intermediate Scale Quantum (NISQ) era (Preskill, 2018), the deployment of learning on quantum devices is becoming an ever-ubiquitous research topic (Cong et al., 2019). Our method can be explored for its implementation in a more quantum-friendly manner beyond the current quantum-inspired one, e.g. by variational quantum circuits (Cerezo et al., 2021), so that it will be more feasible on quantum devices.

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

## A  ANALYSIS OF SPACE TO STORE/TRAIN EMBEDDINGS

**NCE-based NE methods.** In general, there are two most widely adopted objectives used in network embedding (Armandpour et al., 2019; Xiong et al., 2022):

$$\text{NCE-x:} \quad \sum_{(i,j)\in\mathcal{D}} \left[ \log \sigma(\mathbf{x}_i^\top \mathbf{x}_j) + \sum_{k=1}^{K} \mathbb{E}_{j'\sim\mathbb{P}_n} \log \sigma(-\mathbf{x}_i^\top \mathbf{x}_{j'}) \right], \tag{8}$$

and

$$\text{NCE-h:} \quad \sum_{(i,j)\in\mathcal{D}} \left[ \log \sigma(\mathbf{x}_i^\top \mathbf{h}_j) + \sum_{k=1}^{K} \mathbb{E}_{j'\sim\mathbb{P}_n} \log \sigma(-\mathbf{x}_i^\top \mathbf{h}_{j'}) \right]. \tag{9}$$

The NCE-x objective tends to make positively sampled nodes close in the embedding space (when $\mathcal{D}$ is the edge set, it means first-order similarity (Tang et al., 2015)), while the NCE-h objective tends to make nodes with common positive samples close in the embedding space (when $\mathcal{D}$ is the edge set, it means second-order similarity (Tang et al., 2015)). One can see that mathematically NCE-h further utilizes hidden embedding $\mathbf{h}$ compared with NCE-x, and hence NCE-h double the number of training parameters compared with NCE-x. We list the number of parameters of the baseline methods in Table 5.

Table 5: Network embedding methods adopting different types of NCE-based objectives.

| Methods | Objectives | Train Emb. | Store Emb. |
|---|---|---|---|
| VERSE (Tsitsulin et al., 2018) | NCE-x | $|\mathcal{V}|p$ | |
| LINE (Tang et al., 2015) | NCE-x + NCE-h | $1.5|\mathcal{V}|p$ | $|\mathcal{V}|p$ |
| node2vec (Grover & Leskovec, 2016) | NCE-h | $2|\mathcal{V}|p$ | |

**Node2ket.** In this paper, our proposed node2ket adopts NCE-x as the objective, which, however, can be easily substituted with NCE-h by simply doubling the number of TEBs. We list the used parameters of different parts of node2ket in Table 6. Because of its NCE-x core, the numbers of parameters used to train/store embeddings are the same.

Table 6: The number of parameters used to train/store embeddings are the same for node2ket and node2ket+. In the table, 'opt.' is short for 'optional'.

| | TU embeddings (TEBs) trainable | Block weights (opt.) trainable/constants | $r(i,c)$ table constants |
|---|---|---|---|
| Node2ket | $BdC|\mathcal{V}|$ | $B|\mathcal{V}|$ | / |
| Node2ket+ | $Bd\sum_{c=1}^{C} R_c$ | | $C|\mathcal{V}|$ |

## B  RELATIONS TO WORD2KET AND WORD2KETXS

### B.1  UNIFYING WORD2KET AND WORD2KETXS BY TEBS

Here we provide detailed explanation on how to use the language of Tensorized Embedding Blocks to describe word2ket and word2ketXS.

**Word2ket** generates embedding $\mathbf{x}_i$ by:

$$\mathbf{x}_i = \sum_{k=1}^{r} \bigotimes_{j=1}^{n} \mathbf{u}_c^b, \tag{10}$$

where $r$ is the rank and $n$ is the order. Comparing Eq. 10 with the embedding generated by Eq. 3, one can see that in the language of Tensorized Embedding Blocks, the rank is equal to the block number $B$, the order is equal to the column number $C$, with block weights $w_{ib} = 1$ set as constants. Though very similar to the way node2ket generates embeddings, it is worth noting that word2ket

requires that $n = 2^y$ because the embeddings are generated from leaf nodes of a complete binary tree thus computation of final embeddings can be speeded up, while in comparison, the column number $C$ of node2ket can be any integer no less than 2.

**Word2ketXS** generates embedding from linear operators. The embeddings are written as:

$$X = \sum_{k=1}^{r} \bigotimes_{j=1}^{n} U_j^k, \tag{11}$$

where $n$, $r$ has the same meaning as they are in the word2ket, $X$ is the word embedding linear operator which can be understood as the embedding matrix of shape $p \times |\mathcal{V}|$. We write $X = \{\mathbf{x}_i\}_{i=0}^{|\mathcal{V}|-1}$ where $\mathbf{x}_i$ is $i$-th column of $X$ and also the embedding of node $i$. $U_j^k$ is also a linear operator which can be understood as a matrix of shape $q \times t$ ($q^n = p$ and $t^n = |\mathcal{V}|$), and we write it as a set of column vectors $U_j^k = \{\mathbf{u}_{vj}^k\}_{v=0}^{t-1}$. We define $X^k = \{\mathbf{x}_i^k\}_{i=0}^{|\mathcal{V}|-1} = \bigotimes_{j=1}^{n} U_j^k$. By the property of Kronecker product in Lemma 1 whose proof is obvious and omitted, we can find that:

$$\mathbf{x}_{v_1 t^{n-1} + v_2 t^{n-2} + \cdots v_n}^k = \bigotimes_{j=1}^{n} \mathbf{u}_{v_j j}^k, \tag{12}$$

and by $X = \sum_{k=1}^{r} X^b$, we have:

$$\mathbf{x}_{v_1 t^{n-1} + v_2 t^{n-2} + \cdots v_n} = \sum_{k=1}^{r} \bigotimes_{j=1}^{n} \mathbf{u}_{v_j j}^k, \quad v_j = 0, 1, \cdots, t-1. \tag{13}$$

So far we have achieved writing the embedding of word2ketXS in the language of TEBs. It is also worth noting that word2ketXS requires that $n = 2^y$. The reason is the same as word2ket.

**Lemma 1** (**Property of Kronecker Product**). *Given two matrix $A^{p \times q} = \{\mathbf{a}_0, \mathbf{a}_1, \cdots, \mathbf{a}_{q-1}\}$, and $B^{m \times n} = \{\mathbf{b}_0, \mathbf{b}_1, \cdots, \mathbf{b}_{n-1}\}$, suppose $C$ is the Kronecker product of $A$ and $B$, i.e. $C = A \otimes B$, then $C^{pm \times qn} = \{\mathbf{c}_0, \mathbf{c}_1, \cdots, \mathbf{c}_{qn-1}\}$, where $\mathbf{c}_{sm+t} = \mathbf{a}_s \otimes \mathbf{b}_t$.*

### B.2 TECHNICAL DIFFERENCES BETWEEN NODE2KET/NODE2KET+ AND WORD2KET/WORD2KETXS

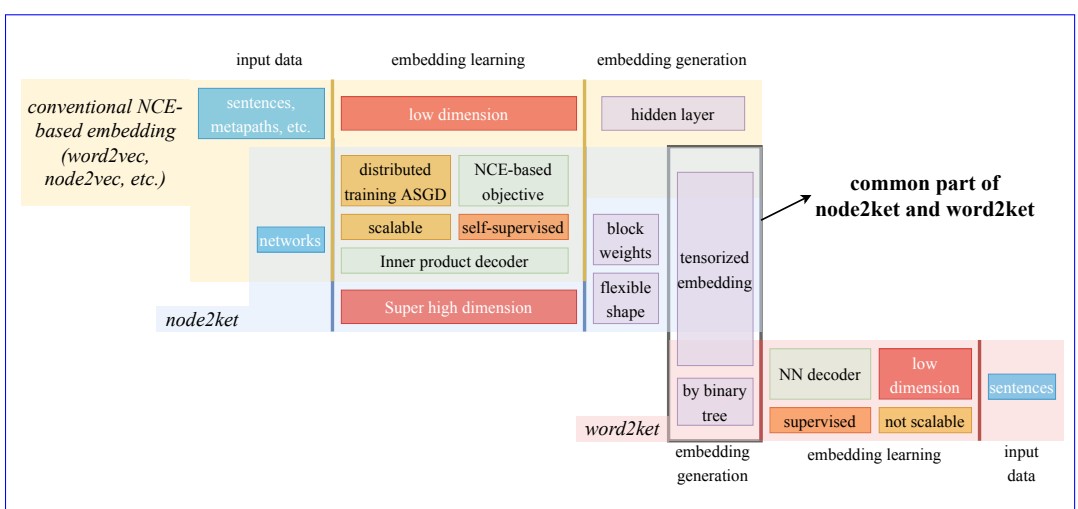

Figure 5: Venn diagram where conventional NCE-based embeddings, node2ket and word2ket are technically compared. We can see that node2ket and word2ket only share common things embedding generation. And indeed our proposed node2ket bridges the gap between conventional embedding methods and tensorized embedding.

The differences are summarized in Table 1 and Fig. 5, which are mainly from the following three aspects:

**i) The embedding generation.** We have shown that both the embedding generation modules of word2ket/word2ketXS are special cases of TEBs in Appendix B.1, and know that the differences of word2ket and word2ketXS lie in the shape of TEBs ($B$, $C$, and $\{R_c\}_{c=1}^C$) and the definition of $r(i,c)$ indices. Then we can find that the differences between the embedding generation models of word2ket/word2ketXS and node2ket/node2ket+ are mainly whether block weights $w_{ib}$ are adopted and whether the shape of TEBs is flexible to be re-designed by re-defining $r(i,c)$. The adoption of block weights is proved to improve model expressiveness when $d$ is small (Fig. 3 b)), and a flexible shape of TEBs is promised to make the embedding model more compressive thus improving space efficiency.

**ii) The embedding learning phase.** Word2ket/word2ketXS train embeddings in a supervised way with an NN decoder, while node2ket/node2ket+ train embeddings by NCE (contrastive learning) which mainly cares about the inner product between the embeddings. In this way, full embeddings are required for word2ket/word2ketXS to feed into the decoder but are not a must for node2ket/node2ket+. Therefore, node2ket/node2ket+ do not need further memory for the full embedding and manage to perform embedding in the Hilbert space of a very large dimension ($2^{32}$ in experiments, which can be indeed more than $2^{128}$ on our machine).

**iii) The input data.** Word2ket accepts sentences as the input data, which is a kind of unstructured data. In comparison, node2ket is designed to process highly structured networks, with an effort on reshaping TEBs to improve both model expressiveness and space efficiency.

## C  TECHNICAL DETAILS

### C.1  GRADIENT DERIVATION

We provide detailed gradient derivation of Eq. 5 and Eq. 6 here. Given two embeddings:

$$\mathbf{x}_i = \sum_{b=1}^B w_{ib} \bigotimes_{c=1}^C \mathbf{u}_{r(i,c)c}^b, \quad \mathbf{x}_j = \sum_{b=1}^B w_{jb} \bigotimes_{c=1}^C \mathbf{u}_{r(j,c)c}^b, \tag{14}$$

we have the inner product:

$$\langle \mathbf{x}_i, \mathbf{x}_j \rangle = \sum_{b}^B \sum_{b'=1}^B w_{ib} w_{jb'} \prod_{c=1}^C \langle \mathbf{u}_{r(i,c)c}^b, \mathbf{u}_{r(j,c)c}^{b'} \rangle, \tag{15}$$

from which we have the partial derivative:

$$\frac{\partial \langle \mathbf{x}_i, \mathbf{x}_j \rangle}{\partial \mathbf{u}_{r(i,c)c}^b} = w_{ib} \sum_{b'=1}^B w_{jb'} \mathbf{u}_{r(j,c)c}^{b'} \prod_{c' \neq c} \langle \mathbf{u}_{r(i,c')c'}^b, \mathbf{u}_{r(j,c')c'}^{b'} \rangle. \tag{16}$$

$$\frac{\partial \langle \mathbf{x}_i, \mathbf{x}_j \rangle}{\partial w_{ib}} = \sum_{b'=1}^B w_{jb'} \prod_{c=1}^C \langle \mathbf{u}_{r(i,c)c}^b, \mathbf{u}_{r(j,c)c}^{b'} \rangle. \tag{17}$$

When $(i,j)$ is a positive sample with label $l = 1$, the objective is $\log \sigma \langle \mathbf{x}_i, \mathbf{x}_j \rangle$, and when it is a negative sample with label $l = 0$, the objective becomes $\log \sigma(-\langle \mathbf{x}_i, \mathbf{x}_j \rangle)$. We define an indicator function $\mathbb{1}_l$ which takes value 1 if $l = 1$ and 0 if $l = 0$, then the objective can be written as

$\log \sigma(2(\mathbb{1}_l - 0.5)\langle \mathbf{x}_i, \mathbf{x}_j \rangle)$. Then we have the following formula with Eq. 16:

$$
\begin{aligned}
&\frac{\partial \log \sigma(2(\mathbb{1}_l - 0.5)\langle \mathbf{x}_i, \mathbf{x}_j \rangle)}{\partial \mathbf{u}^b_{r(i,c)c}} \\
=&\frac{1}{\sigma(2(\mathbb{1}_l - 0.5)\langle \mathbf{x}_i, \mathbf{x}_j \rangle)} \cdot \frac{\partial \sigma(2(\mathbb{1}_l - 0.5)\langle \mathbf{x}_i, \mathbf{x}_j \rangle)}{\partial \mathbf{u}^b_{r(i,c)c}} \\
=&(1 - \sigma(2(\mathbb{1}_l - 0.5)\langle \mathbf{x}_i, \mathbf{x}_j \rangle)) \frac{\partial \big[ 2(\mathbb{1}_l - 0.5)\langle \mathbf{x}_i, \mathbf{x}_j \rangle \big]}{\partial \mathbf{u}^b_{r(i,c)c}} \\
=&\big[ \mathbb{1}_l - \sigma(\langle \mathbf{x}_i, \mathbf{x}_j \rangle) \big] \frac{\partial \langle \mathbf{x}_i, \mathbf{x}_j \rangle}{\partial \mathbf{u}^b_{r(i,c)c}} \\
=&w_{ib} \big[ \mathbb{1}_l - \sigma(\langle \mathbf{x}_i, \mathbf{x}_j \rangle) \big] \sum_{b'=1}^{B} w_{jb'} \mathbf{u}^{b'}_{r(j,c)c} \prod_{c' \neq c} \langle \mathbf{u}^b_{r(i,c')c'}, \mathbf{u}^{b'}_{r(j,c')c'} \rangle,
\end{aligned}
\tag{18}
$$

which is the result of Eq. 5.

For the gradient derivation of block weights $w_{ib}$, Then we have the following formula with Eq. 17:

$$
\begin{aligned}
&\frac{\partial \log \sigma(2(\mathbb{1}_l - 0.5)\langle \mathbf{x}_i, \mathbf{x}_j \rangle)}{\partial w_{ib}} = \big[ \mathbb{1}_l - \sigma(\langle \mathbf{x}_i, \mathbf{x}_j \rangle) \big] \frac{\partial \langle \mathbf{x}_i, \mathbf{x}_j \rangle}{\partial w_{ib}} \\
=&\big[ \mathbb{1}_l - \sigma(\langle \mathbf{x}_i, \mathbf{x}_j \rangle) \big] \sum_{b'=1}^{B} w_{jb'} \prod_{c=1}^{C} \langle \mathbf{u}^b_{r(i,c)c}, \mathbf{u}^{b'}_{r(j,c)c} \rangle,
\end{aligned}
\tag{19}
$$

which is the result of Eq. 6.

## C.2 MODEL LEARNING

We give the pseudo-code of a thread of ASGD in Algorithm 1 and give and illustration of how distributed learning by ASGD is implemented in Fig. 6. The details of embedding regularization and normalization are in Appendix C.2.1.

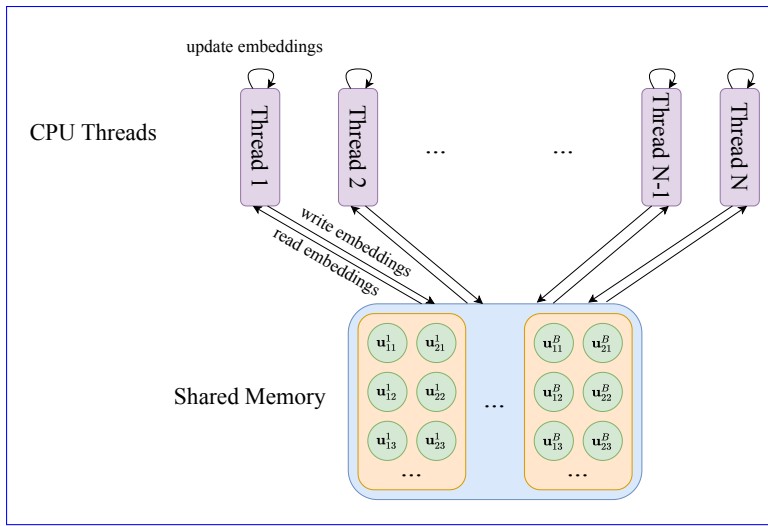

Figure 6: Illustration of how distributed learning by ASGD is performed on embedding learning.

### C.2.1 EMBEDDING REGULARIZATION AND NORMALIZATION FOR TENSOR UNITS

The multi-column architecture of the Tensorized Embedding Blocks brings difficulties to embedding training. Following unexpected results are likely to happen without proper normalization: i) embeddings converge to zero vectors or vectors of infinite length, ii) embeddings converge to the same

---

**Algorithm 1:** A thread of embedding learning via ASGD.

---

1 **Input**: TU embedding dimension $d$, number of negative samples (for each positive sample): $K$, node set $\mathcal{V}$, corpus $\mathcal{D}$, initial learning rate $\eta$ (default 0.025);

  1:  Initialize TU embeddings and block weights.

  2:  **for all** sample a positive sample $(i, j)$ from $\mathcal{D}$ **do**

  3:     Compute the gradients according to Eq. 5 and Eq. 6 (with regularization adopted) with label $l = 1$.

  4:     Update and normalize TU embeddings for node $j$.

  5:     **for** $n = 1, 2, \ldots, K$ **do**

  6:       Sample a negative node $j'$;

  7:       Compute the gradients according to Eq. 5 and Eq. 6 with label $l = 0$.

  8:       Update and normalize TU embeddings for node $j'$.

  9:     **end for**

10:     Update and normalize TU embeddings for node $i$.

11: **end for**

12: **return** TU embedding set $\{\mathbf{u}_{rc}^b\}$ and block weights $\{w_{ib}\}$;

---

vector. We adopts embedding regularization and normalization in the implementation to improve the robustness of the algorithm, as shown in Fig. 7.

**Regularization.** (Fig. 7 a)) Distance penalty terms can be added to positive node pairs to accelerate model convergence and improve the ability to preserve node proximities (Xiong et al., 2022). In the implementation, we adopt Wasserstein-3 distance regularizaton term over TU embeddings of positive node pairs.

**Normalization-1.** (Fig. 7 b)) To increase the stability of embedding training, we make TU embeddings constrained by $\|\mathbf{u}_{rc}\| = 1$. In this way, TU embeddings won't converge to vectors of zero or infinite length.

**Normalization-2.** (Fig. 7 c)) During training, we move the axis towards $-\frac{1}{\sqrt{d}}\mathbf{1}$ for a step. In this way, most of the pairs of embeddings on the unit ball yield a positive inner product. We have known that the inner product of two embeddings $\mathbf{x}_i$ and $\mathbf{x}_j$ is determined by the inner product of the TUs, i.e. $\langle \mathbf{x}_i, \mathbf{x}_j \rangle = \sum_b^B \sum_{b'=1}^B w_{ib} w_{jb'} \prod_{c=1}^C \langle \mathbf{u}_{r(i,c)c}^b, \mathbf{u}_{r(j,c)c}^{b'} \rangle$ in Eq. 15. By making most of inner products between embedding pairs positive, we can mostly avoid the cases where two negative inner products of TUs finally yield a positive inner product of the node embeddings.

a) Regularization      b) Normalization-1      c) Normalization-2

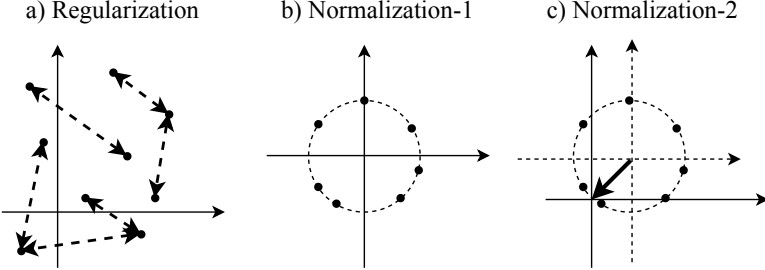

Figure 7: Embedding regularization and normalization.

## D   Proof to Theorems

**Theorem 1.** *Given node set $\mathcal{V}$ and the final embedding dimension $p$, when $r(i, c) = i$ are constants (indicating $R_c = |\mathcal{V}|, \forall 1 \leq c \leq C$) and use block weights as trainable parameters, the total number of parameters is $\#params = B(C|\mathcal{V}|d + |\mathcal{V}|)$. Omitting the integer constraints of $d$, we have:*

$$\min \#params = B|\mathcal{V}|p^{1/\ln p} \ln p + B|\mathcal{V}|.$$

*When the block weights are set as constants, #params = $BC|\mathcal{V}|d$, and we have:*

$$\min \#params = B|\mathcal{V}|p^{1/\ln p}\ln p.$$

*Proof.* The number of parameters for embeddings is $Bd\sum_c R_c = BC|\mathcal{V}|d$, for block weights it is $B|\mathcal{V}|$. Then the total number of training parameters is:

$$\#\text{params} = BC|\mathcal{V}|d + B|\mathcal{V}| \geq B|\mathcal{V}|Cp^{1/C} + B|\mathcal{V}| := f(C), \tag{20}$$

where $\#\text{params} = f(C)$ iff $d = p^{1/C}$. Then we have the derivative:

$$\frac{\mathrm{d}f(C)}{\mathrm{d}C} = B|\mathcal{V}|p^{1/C}(1 - \frac{1}{C}\log p). \tag{21}$$

When $\frac{\mathrm{d}f(C)}{\mathrm{d}C} = 0$, we can reach the solution $C^* = \ln p$. And we have:

$$f(C^*) = B|\mathcal{V}|p^{1/\ln p}\ln p + B|\mathcal{V}| = \min f(C). \tag{22}$$

Now we can reach the conclusion, when $d = p^{1/C}$ and $C = \ln p$:

$$\min \#\text{params} = B|\mathcal{V}|p^{1/\ln p}\ln p + B|\mathcal{V}|. \tag{23}$$

Following the above proof, it is easy to reach the conclusion $\min \#\text{params} = B|\mathcal{V}|p^{1/\ln p}\ln p$ when the block weights are set as constants. $\qquad\square$

**Comments:** By Theorem 1, we prove that we can use $O(p^{1/\ln p}\ln p)$ space for each node embedding in $p$-dimensional space with $B = 1$. For example, when $p = \lfloor e^{1000} \rfloor$ which is too large for classical algorithm to deal with, by node2ket, we may use $C = \ln p = 1000$ and $d = \lceil p^{1/C} \rceil = \lceil e \rceil$, totally about $1000\lceil e \rceil$ or so parameters to train the embedding of dimension $\lfloor e^{1000} \rfloor$ in the Hilbert space.

Before proving Theorem 2, we introduce the following lemma which is obvious by Cauchy inequality:

**Lemma 2.** $\sum_{c=1}^{C} R_c \geq C(\prod_{c=1}^{C} R_c)^{1/C}$, *when the equality holds, $R_1 = R_2 = \cdots = R_C$.*

**Theorem 2.** *Given the node set $\mathcal{V}$ and the final embedding dimension $p$, when node2ket utilizes both the $r(i, c)$ table and block weights, the total number of parameters is $\#params = Bd\sum_{c=1}^{C} R_c + B|\mathcal{V}| + C|\mathcal{V}|$. We have:*

$$\min \#params < BC^*\big(p|\mathcal{V}|\big)^{1/C^*} + B|\mathcal{V}| + C^*|\mathcal{V}|,$$

*where $C^* = \sqrt{\Big[(\ln p|\mathcal{V}|)^2 - \frac{1}{2}\Big(\ln p|\mathcal{V}| + \frac{1}{p|\mathcal{V}|}\Big)\Big]/(|\mathcal{V}|/B + 1)}$. When the block weights are set as constants, we have:*

$$\min \#params < BC^*\big(p|\mathcal{V}|\big)^{1/C^*} + C^*|\mathcal{V}|. \tag{24}$$

*Proof.* By Lemma 2, we have:

$$\begin{aligned}
\#\text{params} &\geq BdC(\prod_{c=1}^{C} R_c)^{1/C} + B|\mathcal{V}| + C|\mathcal{V}| \\
&\geq Bp^{1/C}C(\prod_{c=1}^{C} R_c)^{1/C} + B|\mathcal{V}| + C|\mathcal{V}| \\
&\geq Bp^{1/C}C|\mathcal{V}|^{1/C} + B|\mathcal{V}| + C|\mathcal{V}| \\
&= BC\big(p|\mathcal{V}|\big)^{1/C} + B|\mathcal{V}| + C|\mathcal{V}| := f(C),
\end{aligned} \tag{25}$$

where the inequality on the first line comes from Lemma 2, the second line comes from $d \geq p^{1/C}$, and the third line comes form $\prod_{c=1}^{C} R_c \geq |\mathcal{V}|$. Then we have the derivative:

$$\frac{\mathrm{d}f(C)}{\mathrm{d}C} = B(p|\mathcal{V}|)^{1/C}\Big(1 - \frac{1}{C}\ln p|\mathcal{V}|\Big) + |\mathcal{V}|, \tag{26}$$

for which it is very hard to solve $\frac{\mathrm{d}f(C)}{\mathrm{d}C} = 0$. To give an approximate solution, we first do Taylor Expansion for $(p|\mathcal{V}|)^{1/C}$ as:

$$(p|\mathcal{V}|)^{1/C} = 1 + \frac{1}{C}\ln p|\mathcal{V}| + \frac{1}{2C^2}\Big(\ln p|\mathcal{V}| + \frac{1}{p|\mathcal{V}|}\Big) + O(\frac{1}{C^3}), \tag{27}$$

according to which we have:

$$\begin{aligned}
\frac{\mathrm{d}f(C)}{\mathrm{d}C} &= B\Big[1 + \frac{1}{C}\ln p|\mathcal{V}| + \frac{1}{2C^2}\Big(\ln p|\mathcal{V}| + \frac{1}{p|\mathcal{V}|}\Big) + O(\frac{1}{C^3})\Big]\Big(1 - \frac{1}{C}\ln p|\mathcal{V}|\Big) + |\mathcal{V}| \\
&= B\Big\{1 + \frac{1}{C^2}\Big[\frac{1}{2}\Big(\ln p|\mathcal{V}| + \frac{1}{p|\mathcal{V}|}\Big) - (\ln p|\mathcal{V}|)^2\Big] + O(\frac{1}{C^3})\Big\} + |\mathcal{V}|.
\end{aligned} \tag{28}$$

We let $\frac{\mathrm{d}f(C)}{\mathrm{d}C} \approx 0$ omitting the $O(1/C^3)$ term, then we get a solution:

$$C^* = \sqrt{\Big[(\ln p|\mathcal{V}|)^2 - \frac{1}{2}\Big(\ln p|\mathcal{V}| + \frac{1}{p|\mathcal{V}|}\Big)\Big]/(|\mathcal{V}|/B + 1)}, \tag{29}$$

by which $\frac{\mathrm{d}f(C)}{\mathrm{d}C}\Big|_{C=C^*} \approx 0$. Since $C^*$ is an approximate solution, we have $f(C^*) > \min f(C)$, and hence:

$$\min \#\mathrm{params} < f(C^*), \tag{30}$$

which is the first conclusion of the theorem. For the second one, we just need to remove the second $B|\mathcal{V}|$ term in $f(C)$, similar to the proof of Theorem 1. $\qquad\square$

**Comments:** It is impossible to find analytic solution of $\frac{\mathrm{d}f(C)}{\mathrm{d}C} = 0$, which gives the $C$ that makes $\#params$ minimum. We can only give an upper bound of $\#params$. The procedure to reach Theorem 2 is similar to Theorem 1.

# E    EXPERIMENTS

## E.1    DETAILED EXPLANATION ON THE METRIC PREC@$N$ IN NETWORK RECONSTRUCTION

For large-scale networks, it is infeasible to compute the scores of all $|\mathcal{V}|(|\mathcal{V}|-1)$ node pairs and rank them to reconstruct the raw networks. So, we can only approximately estimate the reconstruction precision by sampling. The description in the paper is actually straight forward. First we sample $|\mathcal{E}|$ randomly matched node pairs as the noise node pairs, denoted as $\mathcal{E}^n$. **Specifically, if we have 10 edges in the given network (i.e. $|\mathcal{E}| = 10$), we randomly sample $10(N-1)$ node pairs as fake (noise) edges $\mathcal{E}^n$ ($|\mathcal{E}^n| = 10(N-1)$).** Then we calculate the scores of edges in $\mathcal{E} \cup \mathcal{E}^n$ and select the top-$|\mathcal{E}|$ pairs as the reconstructed edges. **It means now we have totally $10|\mathcal{E}|$ edges (containing true edges and fake edges), we hope to select 10 edges from the $10N$ edges.** So far, the computation of precision Prec@$N$ becomes obvious. **Since the process contains random sampling, we need to fix the random seed to ensure the fairness of experiments.** Previous works e.g. (Wang et al., 2016) also conduct NR for large-scale networks in this way, which is indeed a common approach in the research.

## E.2    STATISTICS OF DATASETS

The statistics of datasets are given in Table 7. All of them are public online.

Table 7: Statistics and network used in our experiments.

| Dataset | YouTube (resp. Cut) | BlogCatalog | PPI | GR-QC | DBLP |
|---|---|---|---|---|---|
| $|\mathcal{V}|$ | 1,138,499 (resp. 22,579) | 10,312 | 3,890 | 5,242 | 12,591 |
| $|\mathcal{E}|$ | 2,990,443 (resp. 95,596) | 333,983 | 76,584 | 14,496 | 49,627 |
| Avg. degree | 5.25 (resp. 8.46) | 64.78 | 39.37 | 5.53 | 7.88 |
| Type of info. | Social | Social | Biological | Academic | Academic |

### E.3    DETAILED EXPERIMENTAL SETTINGS

### E.3.1    BASELINES

**LINE**[2] (Tang et al., 2015) models first-order and second-order proximity on the adjacency matrix, trains them separately with edge sampling, and concatenates the representations after normalization on node embedding. The number of samples is set as $1 \times 10^8$.

**node2vec**[3] (Grover & Leskovec, 2016) uses biased random walk to explore both breadth-first and depth-first structure, and train node embedding and hidden embedding by negative sampling. We set $p = 1$ and $q = 1$ in the experiments, and window size 10, walk length 80 and number of walks for each node 80.

**VERSE**[4] (Tsitsulin et al., 2018) uses node similarity explicitly by Personalized PageRank (PPR), Adajcency Similarity, and SimRank, and learns node embedding by NCE-x. We use the default version with PPR similarity, damping factor $\alpha = 0.85$. We run $10^5$ epochs for each node.

**NetSMF**[5] (Qiu et al., 2019) obtains node embeddings by sparse matrix factorization on a sparsified dense random walk matrix. We set rank as 512, window size as 10, rounds as 1000, which is the recommended setting by the official code for BlogCatalog.

**ProNE**[6] (Zhang et al., 2019) initializes embedding by sparse matrix factorization, and then enhances the embedding via spectral propagation. We run Python version in the repository with the default setting, i.e. the term number of the Chebyshev expansion $k = 10$, $\theta = 0.5$, and $\mu = 0.2$. We use the enhanced embedding as the node embedding result.

**LouvainNE**[7] (Bhowmick et al., 2020) first adopts Louvain graph partition al-gorithm (Blondel et al., 2008) and then learn node embeddings from the partitions. We use Louvain partition as the graph partition algorithm, and set the damping parameter a=0.01 which is the default value in the repository.

**HHNE**[8] (Zhang et al., 2021) adopts hyperbolic embedding for heterogeneous networks. We feed random walk sequences into the hyperbolic embedding core. We set window size as 5, the number of negative samples as 10. The other parameters follow the default setting of the official repository.

### E.3.2    NODE2KET AND NODE2KET+

**Experimental parameter setting of node2ket in Table 2 (network reconstruction).** On **GR-QC**: we do 10M sampling iterations. On **BlogCatalog**: we do 100M sampling iterations. On **DBLP:** we do 20M sampling iterations. On **PPI**: we do 20M sampling iterations.

**Experimental parameter setting of node2ket+ in Table 2 (network reconstruction).** On **GR-QC**: we do 10M sampling iterations, Louvain graph partition adopts resolution $[800, 1000, 1200, 1400, 1600, 1800, -1, -1]$ (-1 means no parition) for the 8 columns in the TEBs, which results in the row number $[2229, 2495, 2656, 2861, 2978, 3113, 5242, 5242]$. On **BlogCatalog**: we do 100M sampling iterations, Louvain graph partition adopts resolution $[800, 1000, 1200, 1400, 1600, 1800, -1, -1]$ (-1 means no parition) for the 8 columns in the TEBs, which results in the row number $[7122, 7606, 7989, 8296, 8559, 8754, 10312, 10312]$. On **DBLP:** we do 20M sampling iterations, Louvain graph partition adopts resolution $[1000, 1500, 2000, 2500, 3000, -1, -1, -1]$ (-1 means no parition) for the 8 columns in the TEBs, which results in the row number $[4874, 5995, 6944, 7775, 8372, 12591, 12591, 12591]$. On **PPI**: we do 20M sampling iterations, Louvain graph partition adopts resolution $[2000, 2200, 2400, 2600, -1, -1, -1, -1]$ (-1 means no parition) for the 8 columns in the TEBs, which results in the row number $[3372, 3429, 3458, 3505, 3890, 3890, 3890, 3890]$.

---

[2]https://github.com/tangjianpku/LINE

[3]https://github.com/xgfs/node2vec-c

[4]https://github.com/xgfs/verse

[5]https://github.com/xptree/NetSMF

[6]https://github.com/THUDM/ProNE

[7]https://github.com/maxdan94/LouvainNE

[8]https://github.com/ydzhang-stormstout/HHNE

**Experimental parameter setting of node2ket+ in Table 3 (network reconstruction).** We do 1500M sampling iterations, Louvain graph partition adopts resolution $[1500, 3000]$ for the 2 columns in the TEBs, which results in the row number $[81790, 142163]$.

**Experimental parameter setting of node2ket and in Table 4 (link prediction).** On **GR-QC**: we do 100M sampling iterations, we further sample from 2-hop neighbors for positive samples. On **BlogCatalog**: we do 100M sampling iterations. On **DBLP**: we do 50M sampling iterations, we further sample from 3-hop neighbors for positive samples. On **PPI**: we do 100M sampling iterations, we sample from 3-hop neighbors for positive samples.

**Experimental parameter setting of node2ket and in Table 9 and Table 10 (node classification).** On PPI, we do sampling iterations. On YouTube, we do sampling iterations. For baselines we set embedding dimension as $p = 8$. For node2ket, we set $B = 1$, $C = 2$, and $d = 4$, and do 100M sampling iterations.

### E.4 FULL RESULTS OF ABLATION STUDY

The results are give in Table 8. When $B = 1$, $w_{ib}$ becomes meaningless whether $w_{ib}$ are constants or trainable parameters, so the reported results are the same. From the results we see that trainable $w_{ib}$ and the increasing block number $B$ may make a negative influence on the model. It is possibly because the increased number of parameters brings difficulties to model training.

Table 8: Network reconstruction precisions on GR-QC with $d = 8$.

|  | $w_{ib}$ are constants | | | $w_{ib}$ are trainable params | | |
|---|---|---|---|---|---|---|
|  | $C = 2$ | $C = 4$ | $C = 8$ | $C = 2$ | $C = 4$ | $C = 8$ |
| $B = 1$ | 71.32 | 87.04 | 96.20 | 71.32 | 87.04 | 96.20 |
| $B = 2$ | 75.16 | 85.62 | 93.42 | 69.01 | 86.14 | 92.35 |
| $B = 4$ | 71.57 | 79.05 | 89.43 | 72.84 | 83.76 | 89.39 |

### E.5 VISUALIZATION OF RECONSTRUCTION RESULTS

#### E.5.1 INTERPRETABLE VISUALIZATION ON LADDER GRAPH

We conduct node embedding via node2ket on a circular ladder graph of 20 nodes, which is plotted in Fig. 8 a) and the adjacency matrix is visualized in Fig. 8 f). For node2ket, we set $B = 1$, $C = 2$, and $d = 4$. After obtaining the TUs' embeddings, we reconstruct the adjacency matrix from Column 1 and Column 2 of the TEB and visualize them in Fig. 8 b) and c). Then, from the final node embeddings, we compute the reconstructed adjacency matrix and visualize it in real-valued/binarized form as shown in Fig. 8 d) and e). From the results, we have the findings: i) by comparing Fig. 8 b), c), and d), we can see that column 1 and column 2 preserves structures in an approximately orthogonal manner, which means the extracted information is different among columns of a TEB, and ii) by comparing Fig. 8 e) with the ground truth f), node2ket fully reconstructs the input adjacency matrix. To compare the reconstruction performance of different methods on different data, we further investigate a Hypercube graph, 3D-Grid graph, and the real-world Karate Club graph in Appendix E.5. The results directly show the overwhelming performance of node2ket on the network reconstruction task.

#### E.5.2 VISUALIZATION ON OTHER GRAPHS

We investigate a circular ladder graph, a 5-hypercube graph, a 3D-grid graph, and the Karate club graph, and we evaluate all the baselines. The results in Fig. 9 show overwhelming reconstruction ability of node2ket, compared with different methods. We also find that node2ket does not perfectly reconstruct the adjacency matrix for some complex networks, e.g. the 3D-Grid graph and the real-world Karate Club graph.

**Settings.** All baseline methods use 8-dim embedding. For node2ket we set $L = 1$, $C = 2$, and $d = 4$, whose number of parameters for storing embeddings is the same as baselines.

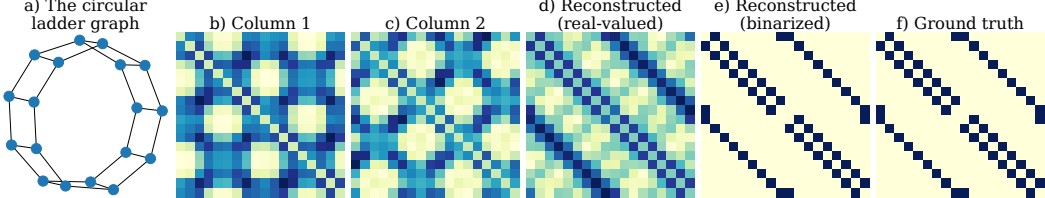

Figure 8: Visualization of the reconstructed adjacency matrix at different levels. A darker color represents a higher value. The subfigure a) is the input circular ladder graph. In b) and c), a color block at the $i$-th row and $j$-th column visualizes the sigmoid score of TUs on the two columns in the TEB, i.e. $\sigma\langle \mathbf{u}_{ic}^1, \mathbf{u}_{jc}^1 \rangle$ mathematically. In d), a color block at the $i$-th row and $j$-th column visualizes the sigmoid score of node embeddings, i.e. $\sigma\langle \mathbf{x}_i, \mathbf{x}_j \rangle$. In e), we binarize the results given by d). And f) is the visualization of the ground-truth adjacency matrix of the graph in a).

### E.6 EXPERIMENTS ON NODE CLASSIFICATION

We conduct experiments on PPI (totally 50 labels) and YouTube (totally 47 labels). After embeddings are obtained, LIBLINEAR (Fan et al., 2008) is used to train one-vs-rest Logistic Regression classifiers. We range the portion of 0training data from 1% to 9% for YouTube and 10% to 90% for PPI. We use Macro-F1 as the metric. In line with (Perozzi et al., 2014; Tang et al., 2015; Wang et al., 2016), for each method we perform 10 times of trials and report the average scores. Results are given in Table 9 and 10, which shows that node2ket consistently outperforms baselines under the setting.

Table 9: Node classification results on PPI.

| Portion of Training Data | 10% | 30% | 50% | 70% | 90% |
|---|---|---|---|---|---|
| line | 8.84 | 9.29 | 9.59 | 10.05 | 9.66 |
| verse | 8.60 | 10.56 | 10.98 | 11.29 | 10.93 |
| node2vec | 6.74 | 9.56 | 10.82 | 11.61 | 11.41 |
| proNE | 7.35 | 8.40 | 8.92 | 9.43 | 9.22 |
| NetSMF | 8.26 | 10.04 | 10.98 | 11.47 | 11.69 |
| LouvainNE | 7.34 | 8.23 | 8.62 | 8.36 | 8.15 |
| HHNE | 9.10 | 10.24 | 10.88 | 10.98 | 10.27 |
| node2ket | **10.53** | **12.23** | **13.15** | **13.56** | **13.01** |

Table 10: Node classification results on YouTube.

| Portion of Training Data | 1% | 3% | 5% | 7% | 9% |
|---|---|---|---|---|---|
| line | 22.74 | 24.97 | 26.40 | 26.99 | 27.44 |
| verse | 24.09 | 26.50 | 27.40 | 28.09 | 28.34 |
| node2vec | 11.54 | 19.44 | 24.14 | 26.69 | 28.37 |
| proNE | 21.41 | 23.09 | 23.54 | 23.77 | 23.82 |
| NetSMF | 25.44 | 27.01 | 27.47 | 27.81 | 27.98 |
| LouvainNE | 20.43 | 21.85 | 21.97 | 22.38 | 22.56 |
| HHNE | 23.42 | 24.82 | 25.26 | 25.90 | 25.91 |
| node2ket | **29.69** | **30.09** | **30.13** | **30.14** | **30.05** |

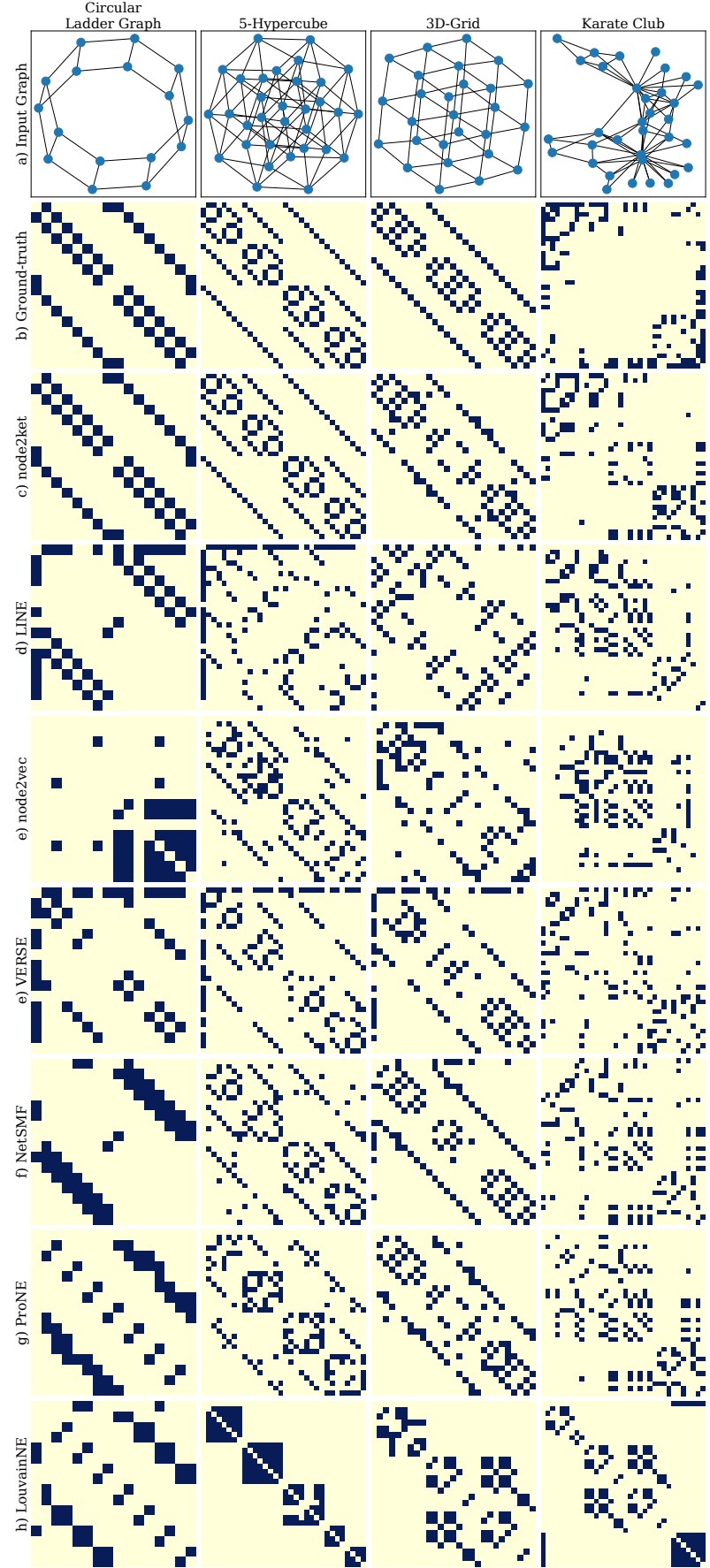

Figure 9: Visualization of reconstruction results of different embedding methods..

