# OpenReview forum: "Quantum-Inspired Tensorized Embedding with Application to Node Representation Learning"
_ICLR.cc/2023/Conference — Submitted to ICLR 2023_

### Official Review · Reviewer_s1h6 · 2022-10-20

**Confidence:** 3
**Correctness:** 3
**Technical Novelty And Significance:** 3
**Empirical Novelty And Significance:** 1
**Recommendation:** 3

**Clarity, Quality, Novelty And Reproducibility:**

The text is sometimes a bit hard to follow due to lacking language, but this is not a huge issue. There are some typos, for example:

- Page 6: Fisrt -> First
- Page 7: abilitty -> ability
- Page 7: in in -> in
- Page 9: reconstruction reconstruction -> reconstruction

There are also multiple parts of the paper that are poorly explained:
- First paragraph in Sec 4.1 is hard to follow.
- In Sec 4.2: "We compute the gradient and update TU embeddings directly without a complete procedure of forward propagation". Do you mean that you are able to avoid using automatic differentiation since you can easily compute the gradients yourself? If so, you should clarify this.
- In Eqs. (5) and (6): You should clarify that this is the gradient of _one_ term in the sum in Eq. (2).
- Theorems 1 and 2: Writing "min # params" is not very clear. It would be better if you wrote in words what the bound is (e.g., "It is sufficient to use x parameters to achieve a generated embedding which is at least p".)
- Fig. 2 caption: "n2k" is an abbreviation which is never defined.
- Toward the end of Sec 4, the part "ii) TEBs are designed to have a flexible ... iii) achieves to fully ... full embedding." These two points are vague. For (ii), it would be much better if you could precisely say which parameters are fixed to what in word2ket/word2ketXS. For (iii), it's not clear what you mean.
- How is Prec@N defined?
- The last 4 sentences in the 1st paragraph of Sec 5.2 ("First we sample ... for different methods.") don't make sense.
- Is the y-axis in Fig. 3 (b) percent or percentage points?


**Strength And Weaknesses:**

Strengths:

S1. The idea itself is interesting. Using compressed tensor formats in ML has been used for a variety of things like analyzing deep NN expressiveness and compression of NN weight layers. I haven't seen it in the context of node embedding before (although my knowledge of the network embedding literature is limited).

S2. Intuitively, it seems like the proposed technique could lead to a substantial reduction in the number of parameters required for a given level of performance in a down-stream task.

Weaknesses:

W1. The main weakness is the experiments. The network reconstruction experiments are not convincing. It is not clear why the ability to reconstruct the original network is important for any interesting down-stream tasks. For example, simply storing the whole network itself only requires 2*(#edges) numbers, and will give perfect "network reconstruction" without any computational cost (so this is a "trivial method" that any other proposal ought to beat). By contrast, the proposed methods node2ket and node2ket+ require many more parameters than are required to store the original network, but give much worse reconstruction accuracy. Based on the provided parameters in the paper and appendix, I calculated the following parameter requirements for the datasets in Table 1.

GR-QC:
- Exact representation: 2*(#edges) = 28,992
- node2ket: 8 * 5242 * 16 = 670,976 (23x more than exact representation)
- node2ket+: (2229 + 2495 + 2656 + 2861 + 2978 + 3113 + 5242 + 5242) * 16 = 429,056 (15x more than exact representation)

BC:
- Exact representation: 667,966
- node2ket: 8 * 10312 * 16 = 1,319,936 (2x more than exact representation)
- node2ket+: (7122 + 7606 + 7989 + 8296 + 8559 + 8754 + 10312 + 10312) * 16 = 1,103,200 (2x more than exact representation)

DBLP:
- Exact representation: 99,254
- node2ket: 8 * 12591 * 16 = 1,611,648 (16x more than exact representation)
- node2ket+: (4874 + 5995 + 6944 + 7775 + 8372 + 12591 + 12591 + 12591) * 16 = 1,147,728 (12x more than exact representation)

PPI:
- Exact representation: 153,168
- node2ket: 8 * 3890 * 16 = 497,920 (3x more than exact representation)
- node2ket+: (3372 + 3429 + 3458 + 3505 + 3890 + 3890 + 3890 + 3890) * 16 = 469,184 (3x more than exact representation)

Similarly, the circular ladder only requires 54 numbers for exact representation, while node2ket uses 64 parameters.

The only case I saw where there was some compression was in Fig 2 (b) when node2vec uses C=2 and d=4. In this case, node2vec uses 0.12 the number of parameters required to store the original network.

The Link prediction experiment in Sec 5.3 is poorly explained. How is the prediction done? What do you mean by "mixture of masked edges and noise edges"? I couldn't find a clarification in the appendix.

**Summary Of The Paper:**

The paper proposes an unsupervised network (graph) embedding technique. Rather than using unstructured vectors to represent nodes, the embedding framework uses sums of Kronecker products of learnable vectors to represent the nodes. This is inspired by entanglement in quantum physics where similar representations are used to represent quantum states.

**Summary Of The Review:**

The method proposed in this paper is an interesting idea, but the experiments are not convincing at all. In particular, it's not clear why network reconstruction matters for down-stream tasks. The paper is also unclear in multiple places; the writing needs to be improved.

---

> ### Author Response · Authors · 2022-11-10
> **Response to Reviewer s1h6**
>
> The reviewer carefully pointed out several typos in the paper and give some important comments. Thanks for the reviewer's hard work and precious suggestions.
>
> The biggest concern of the reviewer comes from network reconstruction. So, in the response, we first explain network reconstruction detailedly. Then we give reasons why node classification is hard to conduct in Hilbert space. Finally, we answer the questions given by the reviewer.
>
> ## On Network Reconstruction
>
> The main concern of the reviewer lies in the experiments, especially the task of network reconstruction (NR). Here we provide more clarification of network reconstruction:
>
> >***Comment 1: Network reconstruction is meaningless for downstream tasks.***
>
> **The significance of preserving network structures in network embedding:** Previous works [1,2] have pointed out that by preserving network structures embedding models can preserve **'first-order' proximity**, which means connected nodes should also be close in the embedding space. For example, in the language of social networks, 'first-order' proximity means that people who are friends also share common interests. And in academic networks, citations usually indicate the relevance of two papers' topics. **In this sense, network reconstruction is highly relevant to downstream tasks. According to Sec. 4.2.5 in [11], 'first-order' proximity preservation will do a favor to label smoothing across the connected nodes.** Researchers have paid much effort into preserving network structures in embedding space since years ago [1,2,3,4]. However, they have never achieved so high precisions as we do (as experiments show in Table 2, 3 and Fig. 9 in the new version of the paper).
>
> **Network reconstruction and link prediction are two complementary tasks:** NR aims to evaluate how well embedding models preserve network structures, which means it is a **'fitting' task**. In comparison, LP is evaluating how well the embedding models predict unknown links, which means that it is a **'predicting' task**. The two topology-based tasks together evaluate the fitting/predicting ability of embedding models.
>
> >***Comment 2: Parameters of embeddings are more than the edgelist, so the method is trivial.***
>
> **Comparing the number of parameters of embeddings and the edgelist does not make sense:** **First,** NR aims to use the embeddings in **continuous vector space** to reconstruct the adjacency matrix which is $O(|\mathcal{V}|^2)$ but not just the edgelist which consumes O(num_edges) space. Comparing in this way would make all embedding methods trivial. **Second,** edgelists cannot be directly applied to other downstream tasks compared with embeddings in the vector space. Embedded networks are much more friendly for downstream task learning. **Furthermore, what we are compressing is not the input data, but the embeddings in high-dimensional Hilbert space.**
>
> **Network reconstruction for real-world networks is a challenging task:** Real-world networks are usually highly non-linear [1,5] hence preserving structures in embedding space is a non-trivial task. It is very hard for existing embedding methods (even those specially designed for preserving structures e.g. LINE-1st[2]) to perfectly achieve that due to the limited dimensions of embedding space (as shown in Table 2, 3 in the new version of the paper).
>
>
>
> ## On Multi-label Node Classification
>
> The reviewer is also interested in other downstream tasks e.g. node classification. **However, node classification is not a trivial task in the Hilbert space:** the embedding dimension of the proposed node2ket reaches 2^32, which is far beyond the processing ability of classical computers/algorithms. It remains an open question for both physicists and computer scientists to study how to implement machine learning in quantum Hilbert space [8,9].
>
> We can only conduct the experiment in a low-dimensional space. **Experiments show that node2ket outperforms all the baselines consistently, even though the representation power of Hilbert space is not fully utilized.** Numerical results are given in the general response.

---

> > ### Author Response · Authors · 2022-11-10
> > **Answers to the Questions**
> >
> > ## Answers to the Questions
> >
> > Here are answers to the questions raised by the reviewer.
> >
> > >***(Q1) First paragraph in Sec 4.1 is hard to follow.***
> >
> > **A1:** In Sec. 4.1, we introduce how tensorized embeddings in Hilbert space are generated by Tensorized Embedding Blocks. We add some introducing words in the new version of the paper.
> >
> > >***(Q2) Meaning of "We compute the gradient and update TU embeddings directly without a complete procedure of forward propagation" in Sec. 4.2.***
> >
> > **A2:** By this sentence, we mean that we do not need automatic differentiation by Pytorch or Tensorflow. We described the details of how to perform ASGD by manually computed gradients in Algorithm 1 and Appendix C.2. It is a very commonly used technique in large-scale NCE-based embeddings including gensim word2vec[7] and node2vec [10]. Since it is not the main concern of the paper, we did not elaborate on how to train embeddings in the main text.
> >
> > >***(Q3) "In Eqs. (5) and (6): You should clarify that this is the gradient of one term in the sum in Eq. (2)."***
> >
> > **A3:** **The gradients given in Eq. 5 and 6 are gradients of all the terms of the objective Eq. 2.** The objective Eq. 2 is given as follows, which has two parts, a positive one and a negative one:
> >
> > $\text{Eq. 2:}\quad \sum\_{(i, j)\in \mathcal{D}} \Big[ \underbrace{\log \sigma (\mathbf{x}\_i^\top \mathbf{x}\_j)}\_{\text{positive samples}} + \underbrace{\sum\_{k=1}^K \mathbb{E}\_{j' \sim \mathbb{P}\_n} \log \sigma (-\mathbf{x}\_i^\top \mathbf{x}\_{j'})}\_{\text{$K$ negative samples for each positive sample}} \Big],$
> >
> > where the embedding $\mathbf{x}\_i$ is given by:
> >
> > $\text{Eq. 3:}\quad \mathbf{x}\_i = \sum\_{b=1}^{B} w\_{ib} \mathbf{x}\_{i}^b, \quad \mathbf{x}\_{i}^b = \bigotimes\_{c=1}^C \mathbf{u}^b\_{r(i,c)c}.$
> >
> > In Eq. 5, we give the gradient $\frac{\partial \log \sigma (2(\mathbb{1}\_{l} - 0.5) \langle \mathbf{x}\_i, \mathbf{x}\_j\rangle)}{\partial \mathbf{u}^{b}\_{r(i,c)c}}$, and in Eq. 6 we give the gradient $\frac{\partial \log \sigma (2(\mathbb{1}\_{l} - 0.5) \langle \mathbf{x}\_i, \mathbf{x}\_j\rangle)}{\partial w\_{ib}}$. We can see that by setting label $\mathbb{1}\_l$ to 0 (negative ) or 1 (positive), we can obtain the gradients w.r.t. all elements $\mathbf{u}\_{r(i,c)c}^b$ (by Eq. 5) and $w\_{ib}$ (by Eq. 6).
> >
> > >***(Q4) Theorems 1 and 2: Writing “min # params” is not very clear.***
> >
> > **A4:** Thanks for your advice. It means the minimum number of used parameters to store embeddings. We add more interpretation of the theorems in both the main text and Appendix D.
> >
> > >***(Q5) Fig. 2 caption: “n2k” is an abbreviation which is never defined.***
> >
> > **A5:** 'n2k' has been defined formally in the first line of Sec. 4.3 in the old version of the paper.
> >
> > >***(Q6) The description of the differences between our methods and word2ket in Sec 4.5 is confusing: "Toward the end of Sec 4, the part “ii) TEBs are designed to have a flexible … iii) achieves to fully … full embedding.” These two points are vague. For (ii), it would be much better if you could precisely say which parameters are fixed to what in word2ket/word2ketXS. For (iii), it’s not clear what you mean."***
> >
> > **A6:** For page limit, we have detailedly discussed the differences between word2ket and our node2ket in Appendix B. **Please refer to the general response where we highlight the differences.** In the new version of the paper, we move the part in the appendix to the main text in Sec. 4.5.

---

> > > ### Author Response · Authors · 2022-11-10
> > > **Answers to the Questions (Cont.)**
> > >
> > > >***(Q7&8) How is Prec@N defined? The description of preparing datasets for network reconstruction in Sec. 5.2 is confusing.***
> > >
> > > **A7&8:** We have added more description words in Appendix E.1 to detailedly explain the metric Prec@N. For large-scale networks, it is infeasible to compute the scores of all $O(|\mathcal{V}|^2)$ node pairs and rank them to reconstruct the raw networks. So, we can only approximately estimate the reconstruction precision by sampling. **The description in the paper is actually straightforward.** "First we sample $(N-1)|\mathcal{E}|$ randomly matched node pairs as the noise node pairs, denoted as $\mathcal{E}^n$." **That means if we have 10 edges in the given network ($|\mathcal{E}|=10$), we randomly sample $10(N-1)$ node pairs as fake (noise) edges $\mathcal{E}^n$.** "Then we calculate the scores of edges in $\mathcal{E}^n \cup \mathcal{E}$ and select the top-$|\mathcal{E}|$ pairs as the reconstructed edges." **That means now we have $10N$ edges (containing true edges and fake edges), and we hope to select top-10 edges from the $10N$ edges.** "Then the precision is calculated in the same way as the former case." **Here the computation of precision, i.e. Prec@N, becomes obvious by computing the portion of true edges in the edges of top-$|\mathcal{E}|$ scores.** "For a fair comparison, we fix the random seed the same in the evaluation for different methods." **That means, since the process contains random sampling, we need to fix the random seed to ensure the fairness of experiments.** Previous works e.g. [1] also conduct NR for large-scale networks in this way, which is a common approach in the research.
> > >
> > > >***(Q9) Is the y-axis in Fig. 3 (b) percent or percentage points?***
> > >
> > > **A9:** Here we mean percentage points. Thanks for your advice. We have modified that in the new version of the paper.
> > >
> > > -----
> > >
> > > Ref:
> > >
> > > [1] Structural Deep Network Embedding. Wang et al. KDD 16.
> > >
> > > [2] LINE: Large-scale Information Network Embedding. Tang et al. WWW 15.
> > >
> > > [3] Embedding Heterogeneous Information Network in Hyperbolic Space. Zhang et al. TKDD 21.
> > >
> > > [4] VERSE: Versatile Graph Embeddings from Similarity Measures. Tsitsuli et al. WWW 18.
> > >
> > > [5] Learning deep architectures for ai. Bengio et al. Foundations and trend in Machine Learning 2009.
> > >
> > > [6] https://radimrehurek.com/gensim/
> > >
> > > [7] Distributed representations of words and phrases and their compositionality. Mikolov et al. NIPS 2013.
> > >
> > > [8] Supervised learning with quantum-enhanced feature spaces. Havlíček V, Córcoles A D, Temme K, et al. Nature, 2019.
> > >
> > > [9] Machine learning of high dimensional data on a noisy quantum processor. Peters E, Caldeira J, Ho A, et al. npj Quantum Information, 2021.
> > >
> > > [10] node2vec: Scalable Feature Learning for Networks. Grover et al. KDD 16.
> > >
> > > [11] Learning Regularized Noise Contrastive Estimation for Robust Network Embedding. Xiong et al. TKDE 2022.

---

> > ### Comment · Reviewer_s1h6 · 2022-11-10
> > **Still not convinced of utility of method**
> >
> > Thank you for the response.
> >
> > I think it would be fine to have a method which produces embeddings that have more parameters than the original network if the embeddings clearly improve performance in down-stream tasks. My issue with the paper is that network reconstruction is emphasized so heavily, while there is just a single experiment on a down-stream task. Since the embedding requires more storage than the network itself, representation of the network clearly is not an actual application. The paper would be more convincing if clear advantages were shown in a range of down-stream tasks.
> >
> > With this in mind, I leave my score as it is.

---

> > > ### Author Response · Authors · 2022-11-11
> > > **Response to Reviewer s1h6**
> > >
> > > Dear reviewer,
> > >
> > > Network reconstruction, node classification, and link prediction are the three most popular tasks in the literature on single-network embedding [1,2,3,4,5,6,7] (indeed for almost all works in the area). We have provided all the experiment results with newly added results on node classification in our revision, and our methods (almost) consistently achieve the best accuracy.
> > >
> > > We are looking forward to your feedback on our specific new comparison upon your request.
> > >
> > > **For your mentioned "a range of tasks", could you please give another specific example of downstream tasks for single-network embedding? We will try our best and seize this opportunity to conduct the experiments as our follow-up. Thank you!**
> > >
> > > ## Network Reconstruction
> > >
> > > The precisions of network reconstructions are given as below:
> > >
> > > | dataset | node2ket  | node2ket+ | LINE  | node2vec | VERSE | NetSMF | ProNE | LouvainNE | HHNE  |
> > > | ------- | --------- | --------- | ----- | -------- | ----- | ------ | ----- | --------- | ----- |
> > > | GR-QC   | **97.31** | 92.43     | 54.19 | 66.38    | 56.08 | 30.26  | 56.37 | 43.47     | 50.50 |
> > > | BC      | **26.07** | 25.06     | 4.80  | 21.16    | 10.96 | 7.64   | 10.10 | 6.79      | 0.69  |
> > > | DBLP    | 86.40     | **86.62** | 2.34  | 21.17    | 16.96 | 18.08  | 17.76 | 21.33     | 11.40 |
> > > | PPI     | 74.07     | **82.25** | 15.18 | 48.13    | 27.59 | 17.14  | 24.02 | 17.77     | 18.23 |
> > >
> > >
> > > | Method        | Prec@2    | Prec@10   | Prec@50   | Running Time (s) |
> > > | ------------- | --------- | --------- | --------- | ---------------- |
> > > | LINE          | 71.35     | 36.54     | 16.92     | 502.67           |
> > > | node2vec      | 44.85     | 9.80      | 3.14      | 3979.52          |
> > > | VERSE         | 53.68     | 34.20     | 25.08     | 435.03           |
> > > | ProNE         | 76.80     | 41.79     | 18.96     | 85.71            |
> > > | LouvainNE     | 85.74     | 70.89     | **63.12** | 8.63             |
> > > | **node2ket+** | **87.51** | **74.75** | 61.69     | 122.81           |
> > >
> > >
> > > ## Node Classification
> > >
> > > The Macro-F1 score on YouTube and PPI are given as below:
> > >
> > > | **YouTube**  | 1%        | 3%        | 5%        | 7%        | 9%        |
> > > | ------------ | --------- | --------- | --------- | --------- | --------- |
> > > | LINE         | 22.74     | 24.97     | 26.4      | 26.99     | 27.44     |
> > > | node2vec     | 11.54     | 19.44     | 24.14     | 26.69     | 28.37     |
> > > | VERSE        | 24.09     | 26.50     | 27.40     | 28.09     | 28.34     |
> > > | ProNE        | 21.41     | 23.09     | 23.54     | 23.77     | 23.82     |
> > > | NetSMF       | 25.44     | 27.01     | 27.47     | 27.81     | 27.98     |
> > > | LouvainNE    | 20.43     | 21.85     | 21.97     | 22.38     | 22.56     |
> > > | **HHNE**  | 23.42     | 24.82     | 25.26     | 25.90     | 25.91     |
> > > | **node2ket** | **29.69** | **30.09** | **30.13** | **30.14** | **30.05** |
> > >
> > >
> > > | **PPI**      | 10%       | 30%       | 50%       | 70%       | 90%       |
> > > | ------------ | --------- | --------- | --------- | --------- | --------- |
> > > | LINE         | 8.84      | 9.29      | 9.59      | 10.05     | 9.66      |
> > > | node2vec     | 6.74      | 9.56      | 10.82     | 11.61     | 11.41     |
> > > | VERSE        | 8.60      | 10.56     | 10.98     | 11.29     | 10.93     |
> > > | ProNE        | 7.35      | 8.40      | 8.92      | 9.43      | 9.22      |
> > > | NetSMF       | 8.26      | 10.04     | 10.98     | 11.47     | 11.69     |
> > > | LouvainNE    | 7.34      | 8.23      | 8.62      | 8.36      | 8.15      |
> > > | **HHNE**  | 9.10      | 10.24     | 10.88     | 10.98     | 10.27     |
> > > | **node2ket** | **10.53** | **12.23** | **13.15** | **13.56** | **13.01** |
> > >
> > > ## Link Prediction
> > >
> > > The link prediction precisions are given as below:
> > >
> > > | Datasets | node2ket  | LINE  | node2vec | VERSE | NetSMF | ProNE | LouvainNE | HHNE  |
> > > | -------- | --------- | ----- | -------- | ----- | ------ | ----- | --------- | ----- |
> > > | GR-QC    | **97.24** | 95.86 | 91.72    | 93.79 | 87.59  | 93.79 | 86.90     | 93.10 |
> > > | BC       | **81.95** | 53.08 | 77.58    | 33.35 | 59.34  | 75.27 | 57.99     | 38.77 |
> > > | DBLP     | **91.95** | 75.45 | 82.09    | 68.81 | 82.09  | 85.31 | 73.44     | 85.92 |
> > > | PPI      | **79.12** | 55.15 | 77.58    | 51.55 | 72.16  | 73.97 | 57.99     | 65.72 |
> > >
> > > ----
> > > Ref:
> > >
> > > [1] Structural Deep Network Embedding. Wang et al. KDD 16.
> > >
> > > [2] LINE: Large-scale Information Network Embedding. Tang et al. WWW 15.
> > >
> > > [3] VERSE: Versatile Graph Embeddings from Similarity Measures. Tsitsuli et al. WWW 18.
> > >
> > > [4] node2vec: Scalable Feature Learning for Networks. Grover et al. KDD 16.
> > >
> > > [5] DeepWalk: Online Learning of Social Representations. Perozzi et al. KDD 14.
> > >
> > > [6] GraRep: Learning Graph Representations with Global Structural Information. Cao et al. CIKM 15.
> > >
> > > [7] NetSMF: Large-Scale Network Embedding As Sparse Matrix Factorization. Qiu et al. WWW 19.

---

### Official Review · Reviewer_vWD3 · 2022-10-20

**Confidence:** 4
**Correctness:** 4
**Technical Novelty And Significance:** 3
**Empirical Novelty And Significance:** 3
**Recommendation:** 8

**Clarity, Quality, Novelty And Reproducibility:**

**Clarity and Quality:**

The full paper is quite well-written and clearly introduces the main contributions of the work. The clarity and quality of the paper can be improved if authors pay some attention to the following several points:

1.
2. The clarity of the “objective function” part can be further improved. As one of the readers (or reviewers) of this paper, I’m not familiar with network embedding well. So when I went through the “objective function” part, I got confused with some vocabulary, such as what “negative sample” and the distribution $\mathbb{P}_n$ stand for in this scenario? and why in the second term of Eq. (2) there is no index $k$ in the summation over $k\in{}[K]$? Maybe this knowledge is well-known by network analyzers, but it would be more friendly for general ML researchers if this part could be introduced more clearly.
3. In Sec 4.3 the paper discussed the theoretically optimal to compressive power (although I fully understand what the meaning of “compressive power” is). It would be better if more explanations (such as insights) could be given following each theoretical result. For example, why does the logarithm appear in the equations? If the assumptions are plausible in practice? and so on.

**Novelty:**

Although this might not be the first paper to apply tensors to network embedding, the proposed methods and the used tricks are very interesting and inspireful for tensor researchers such as the graph partition used in NODE2KET+.

**Reproducibility**

The experimental settings are clearly introduced.

**Strength And Weaknesses:**

++Strength

1. Very interesting model and the combination with quantum is also very smooth and clearly discussed.
2. The improvement in the empirical performance is remarkable.

— Weakness

1. The theoretical results are not well explained.

**Summary Of The Paper:**

This paper studied a novel network embedding method inspired by quantum fields and modeled with tensors and contractions (i.e., outer product). More specially, the embedding vector for each node is represented by a linear combination of rank-one tensors, equivalent to the CP model studied in numerical algebra. In the paper, the authors also discussed the compressive power of the proposed method in a theoretical way. In the experiment part, this paper showed superior performance on network analysis tasks and evaluated the parallelizability and interpretability, which is crucial in practice.

**Summary Of The Review:**

This is excellent work at least in the tensor community. The problem focused in this paper would inspire lots for tensor researchers to explore more valuable applications in machine learning. It also gives a bridge to connect machine learning to quantum computing.

---

> ### Author Response · Authors · 2022-11-10
> **Response to Reviewer vWD3**
>
> Thanks for your positive comments and valuable suggestions! We are glad to see that you think our work is interesting and that the results are remarkable.
>
> Here are the answers to your questions.
>
> - **Q1: What does 'negative samples' mean?**
>
> A1: The embedding paradigm is based on noise contrastive estimation, which divides node pairs as positive ones and negative ones. Positive node pairs can just be the links in the networks. Negative node pairs are usually by random sampling: given a node $i$, randomly sample a node $j'$ from the network as the negative sample. Positive and negative samples are partly overlapped, but it won't affect model performance due to the sparsity of real-world networks. We add some notes to Eq. 2 for better understanding.
>
> - **Q2: Why no index $k$ are in the summation over?**
>
> A2: $K$ is the number of negative samples, which means we perform $K$ times of negative sampling. So $k$ would not show in the summation term.
>
> - **Q3: Why does logarithm appear in the equations? If the assumptions are plausible in practice?**
>
> A3: Take Theorem 1 as an example, as proved in Appendix D, when $C=\ln p$, the number of used parameters is the minimum. **In practice, the parameters $C$ and $d$ are constrained to be integers. Hence the actually used parameters are indeed slightly more than we proved in the theorems.** We have also made this point clear in the proofs in Appendix D. Assuming that we are conducting node2ket in a $2^{10}$-dimensional space ($p=2^{10}=1024$), then we may choose $C=\lceil 10 \ln 2\rceil = 7$ and $d=\lceil p^{1/C} \rceil = 3$, which probably consumes the least local space. In this case, we use $Cd=21$ parameters to represent a 1024-dim embedding vector. When it comes to Theorem 2, the procedure is similar.
>
> **The two assumptions are plausible, and won't be affected by the integer constraints.** In assumption 1, we assume that the dimension of generated embedding $d^C$ should not be smaller than $p$. In assumption 2, we assume that the number of generated node embeddings should be no smaller than the number of nodes. Both are plausible.
>
> We have made some modifications in the new version of the paper according to the review, including:
> - (Q1&2) Add more description words in the background of NCE-based network embedding in Sec. 3.1.
> - (Q3) Add comments and examples to explain the theorems in Appendix D.

---

### Official Review · Reviewer_koXd · 2022-10-27

**Confidence:** 2
**Correctness:** 3
**Technical Novelty And Significance:** 1
**Empirical Novelty And Significance:** Not applicable
**Recommendation:** 3

**Clarity, Quality, Novelty And Reproducibility:**

Both novelty and clarity need to be improved.


**Strength And Weaknesses:**

Strength
- It is interesting to employ tensorized embedding to network embedding field.

Weakness
- The novelty is very limited. Most of the parts are similar to word2ket. Thus, the contribution to network embedding field is very limited, since only the objective function is different.
- There is no contribution on the graph topology exploitation. The topology is only used in the the objective function, which is the same as pervious network embedding ones.
- The experimental evaluations are not sufficient. The performance on node representation is only verified on topology-related tasks. However, that on node classification task is not given. Furthermore, the performance on topology-based tasks are only compared with the classic methods, instead of the SOTA such as hyperbolic space embedding methods.
- The writing is redundant. Most content is similar to word2ket.


**Summary Of The Paper:**

This paper employs the tensorized embedding proposed in word2ket to the network embedding. It is similar with the second order part in network embedding method LINE, which is motivated from word2vec. Its superiority is verified on the tasks of topology reconstruction and link prediction.

**Summary Of The Review:**

My main concern are the limited novelty and contribution and insufficient evaluations show in weakness.

---

> ### Author Response · Authors · 2022-11-10
> **Response to Reviewer koXd**
>
> Thanks for the reviewer's comments and good suggestions. We are glad that the reviewer thinks our paper is interesting to the embedding community. However, reviewers seem to have some misunderstandings about novelty and experiments. Here are our responses to the reviewer's comments.
>
> >***Point i) "It is similar with the second order part in network embedding method LINE."***
>
> **Answer i). On the novelty compared with conventional embedding.** Our proposed node2ket can be applied on any NCE-based network embedding methods including LINE-1st/2nd, node2vec, verse, struc2vec, and so on. Different from previous researches that work on how to design positive/negative samples for NCE-based embedding core, **we propose a brand new core for embedding learning in Hilbert space.** **Indeed, we are the first to propose and successfully implement tensorized embedding in quantum Hilbert space on large-scale networks based on NCE.**
>
> >***Point ii). "Most of the parts are similar to word2ket. Thus, the contribution to network embedding field is very limited, since only the objective function is different.".***
>
> **Answer ii). On the novelty compared with word2ket.** Generally speaking, a complete network embedding procedure basically has three steps: 1) prepare the input data, 2) generate embedding for the input data, and 3) learn the embeddings (loss function design and training techniques). **The only common part of node2ket and word2ket is in 2) that the tensorized embedding model of word2ket/word2ketXS are special cases of our proposed embedding module Tensorized Embedding Blocks (TEBs).**
>
>
> **For both points i) and ii).** We have made the differences between node2ket, conventional embedding methods, and word2ket clear in Sec. 4.5 and Fig. 5 in Appendix B. **Indeed, we bridge the gap between conventional embedding methods and tensorized embedding word2ket. Please check Sec. 4.5 and Fig. 5 in Appendix B, and our explanation in the general response carefully.**
>
> >***Point iii). "There is no contribution on the graph topology exploitation."***
>
> **Answer iii).** Topology exploitation is not the main concern of this paper. **We propose a brand new embedding core in this paper, by which we use the least information (only the raw structure of networks) to achieve the best performance.**  We leave topology exploitation with our embedding core to further improve the performance as future work.
>
> >***Point iv). Requirement for hyperbolic embedding baselines and extra experiments on node classification.***
>
> **Answer iv). New baseline HHNE [1]:** We provide the experimental results of HHNE (a SOTA hyperbolic network embedding baseline) for network reconstruction (NR) and link prediction (LP) where the precisions are given in the table below. **We can see that there is still a huge gap between HHNE and node2ket on the two tasks.** Hyperbolic embedding is still a conventional one in a low-dimensional embedding space $R^p$ compared with node2ket which is in the super high-dimensional **quantum** Hilbert embedding space $R^{d^C}$.
> **Node classification is not a trivial task in the Hilbert space:** the embedding dimension of the proposed node2ket reaches 2^32, which is far beyond the processing ability of classical computers/algorithms. It remains an open question for both physicists and computer scientists to study how to implement machine learning in quantum Hilbert space [2,3]. Due to the difficulty, we can only conduct experiments in a low-dimensional embedding space. Numerical results are given in the general response, **which show that node2ket outperforms all the baselines consistently, even in a low-dimensional embedding space where the representation power of Hilbert space is not fully utilized.**
>
>
> | Network Reconstruction | BlogCatalog | PPI       | GR-QC     | DBLP      |
> | ---------------------- | ----------- | --------- | --------- | --------- |
> | HHNE                   | 0.69        | 18.23     | 50.50     | 11.40     |
> | node2ket               | **26.07**   | **74.07** | **97.31** | **86.40** |
>
> | Link Prediction | BlogCatalog | PPI       | GR-QC     | DBLP      |
> | --------------- | ----------- | --------- | --------- | --------- |
> | HHNE            | 38.77       | 65.72     | 93.10     | 85.92     |
> | node2ket        | **81.95**   | **79.12** | **97.24** | **91.95** |
>
> >***Point v) The writing is redundant.***
>
> **Answer v).** Thanks for your advice, we have paid much more effort into paper writing and we submit a new version of the paper.
>
> -----------
>
> Ref:
>
> [1] Embedding Heterogeneous Information Network in Hyperbolic Space. Zhang et al. TKDD 2021.
>
> [2] Supervised learning with quantum-enhanced feature spaces. Havlíček V, Córcoles A D, Temme K, et al. Nature, 2019.
>
> [3] Machine learning of high dimensional data on a noisy quantum processor. Peters E, Caldeira J, Ho A, et al. npj Quantum Information, 2021.

---

### Author Response · Authors · 2022-11-10
**General Response to All Reviewers**

# General Response to All the Reviewers

We are wholeheartedly thankful for the reviewers' reviews of this paper and their valuable suggestions. However, because our embedding model is quantum-inspired, and the efficient learning and practical application on large-scale datasets utilize techniques of distributed training. Fully understanding the paper may require a strong knowledge of network embedding, distributed training, and quantum mechanics. They cannot be detailedly introduced from scratch especially in 9-page main text, so we have to put technical details in Appendix (referred to in the main text), which, however, might be easily omitted by the reviewers. That may be the reason for the misunderstandings among the reviewers.

In summary, reviewers are mostly concerned about: i) our novelty (including what we contribute to the embedding community and the differences between our methods and word2ket), and ii) experiments on node classification. In our following response, first, we give a retrospect on our main contributions. Second, we discuss the differences between our methods and word2ket in detail. And then we add experimental results on node classification. Finally, we summarize the main revisions in our new version of the paper.

## Our Contributions

- **The embedding generation module unifies previous conventional/tensorized embedding methods.** The proposed Tensorized Embedding Blocks (TEBs) generate embeddings in Hilbert space by tensor product and entanglement. Conventional embedding lookup tables and tensorized embeddings by word2ket/word2ketXS are both special cases of our framework. Detailed proofs are given in Sec. 4.5 and Appendix B.
- To the best of our knowledge, **we are the first to propose and successfully implement tensorized embedding in quantum Hilbert space on large-scale networks based on Noise Contrastive Estimation (NCE)**. The algorithm design and implementation require much domain knowledge and experience in distributed training and are non-trivial. In this sense, we bridge the gap between tensorized embedding models and conventional NCE-based embedding models.
- **Overwhelming experimental results on network reconstruction and link prediction (plus node classification in the new version of the paper).** Notably, the network reconstruction precision on DBLP is more than 3 times higher than the best one of baselines LouvainNE. **The scalability and parallelizability are also proved in experiments.**


## We Are Very Different from Word2ket

We have detailedly compared our methods and word2ket in Appendix B in the initial version of the submission. Table 6 (it becomes Table 1 in the new version of the paper) and Figure 5 summarize the differences. First of all, a complete network embedding procedure basically has three steps: i) prepare the input data, ii) generate embedding for the input data, and iii) learn the embeddings. Only in the embedding generation part word2ket and our node2ket share common things.

- For embedding generation. The embedding generation procedures of word2ket/word2ketXS are special cases of our proposed Tensorized Embedding Blocks (TEBs) (see Appendix B.1 for proof). Compared to word2ket/word2ketXS, TEBs can generate tensorized embeddings of a more flexible shape and have a stronger representation capability by utilizing block weights.
- For the input data. Word2ket/word2ketXS accept sentences as the input, while node2ket accepts highly-structured data networks as the input. **Node2ket is specially designed for structured data by reshaping TEBs and designing the $r(i,c)$ table.** For example, in node2ket+, we encode the structure of input data into TEBs by setting $r(i,c)=r(j,c)$ for node $i$ and $j$ that belong to the same community.
- For embedding learning. Word2ket/word2ketXS train embeddings by **a supervised NN decoder**, which consumes further space consuming and requires computing the full embeddings (whose space is $O(d^C)$ for each word). As a result, it only generates embeddings in **at most 8000-dimensional space**. In comparison, node2ket train embeddings self-supervisedly by an inner-product based objective, which is much lighter. Consequently, node2ket does not require computing full embeddings, which means node2ket only consumes $O(Cd)$ space for each node. In the experiment **the embedding dimension of node2ket reaches $2^{32}$**. **The super high dimensional Hilbert space endows node2ket with super expressiveness.**

We have also detailedly discussed the advantages of node2ket compared with word2ket in our new version of the paper from the above three aspects.

---

> ### Author Response · Authors · 2022-11-10
> **Experiments on Node Classification and Main Revisions in the New Version of the Paper**
>
> ## Experiments on Node Classification
>
> **The reason we did not do experiments on node classification in the initial version of the paper:** Reviewer 1 and 3 both mention the experiments of node classification. **However, node classification is not a trivial task in the Hilbert space:** the embedding dimension of the proposed node2ket reaches 2^32, which is far beyond the processing ability of classical computers/algorithms. It remains an open question for both physicists and computer scientists to study how to implement machine learning in quantum Hilbert space [1,2]. In this sense, node classification may not be a good choice for downstream tasks. **The task of link prediction is a better choice than node classification to evaluate model performance in Hilbert space since the computation can be simplified in the super high-dimensional Hilbert space (see Eq. 15 in Appendix C.1).**
>
> Due to the difficulty, we can only conduct experiments in a low-dimensional embedding space. Here we provide the classification results (Macro-F1) on PPI and YouTube. We also add the results in Appendix E.5. **Results show that node2ket consistently outperforms baselines.**
>
> | **YouTube**  | 1%        | 3%        | 5%        | 7%        | 9%        |
> | ------------ | --------- | --------- | --------- | --------- | --------- |
> | LINE         | 22.74     | 24.97     | 26.4      | 26.99     | 27.44     |
> | node2vec     | 11.54     | 19.44     | 24.14     | 26.69     | 28.37     |
> | VERSE        | 24.09     | 26.50     | 27.40     | 28.09     | 28.34     |
> | ProNE        | 21.41     | 23.09     | 23.54     | 23.77     | 23.82     |
> | NetSMF       | 25.44     | 27.01     | 27.47     | 27.81     | 27.98     |
> | LouvainNE    | 20.43     | 21.85     | 21.97     | 22.38     | 22.56     |
> | **HHNE** [3] | 23.42     | 24.82     | 25.26     | 25.90     | 25.91     |
> | **node2ket** | **29.69** | **30.09** | **30.13** | **30.14** | **30.05** |
>
>
> | **PPI**      | 10%       | 30%       | 50%       | 70%       | 90%       |
> | ------------ | --------- | --------- | --------- | --------- | --------- |
> | LINE         | 8.84      | 9.29      | 9.59      | 10.05     | 9.66      |
> | node2vec     | 6.74      | 9.56      | 10.82     | 11.61     | 11.41     |
> | VERSE        | 8.60      | 10.56     | 10.98     | 11.29     | 10.93     |
> | ProNE        | 7.35      | 8.40      | 8.92      | 9.43      | 9.22      |
> | NetSMF       | 8.26      | 10.04     | 10.98     | 11.47     | 11.69     |
> | LouvainNE    | 7.34      | 8.23      | 8.62      | 8.36      | 8.15      |
> | **HHNE** [3] | 9.10      | 10.24     | 10.88     | 10.98     | 10.27     |
> | **node2ket** | **10.53** | **12.23** | **13.15** | **13.56** | **13.01** |
>
>
>
>
> Ref:
>
> [1] Supervised learning with quantum-enhanced feature spaces. Havlíček V, Córcoles A D, Temme K, et al. Nature, 2019.
>
> [2] Machine learning of high dimensional data on a noisy quantum processor. Peters E, Caldeira J, Ho A, et al. npj Quantum Information, 2021.
>
> [3] Embedding Heterogeneous Information Network in Hyperbolic Space. Zhang et al. TKDD 2021.
>
>
> ## Main Revisions in the New Version of the Paper
>
> We highlight the revised parts in blue in the new version of the paper.
>
> - (Reviewers 1(koXd) & 3 (s1h6)) We add a Venn Diagram in Fig. 5 to further visualize the differences between conventional embeddings, word2ket, and node2ket.
> - (Reviewer 1(koXd) & 3 (s1h6)) We move the analysis of the differences between word2ket and node2ket to the main text in Sec. 4.5, and summarize the difference in Table 1 in the new version of the paper.
> - (Reviewers 1(koXd) & 3 (s1h6)) We add a section of experimental results on node classification in Appendix E.5, which shows that our method consistently outperforms baselines.
> - (Reviewer 1 (koXd)) We add experimental results of HHNE.
> - (Reviewer 3 (s1h6)) We fix the typos.
> - (Reviewer 3 (s1h6)) We add Fig. 6 to illustrate ASGD distributed training for the NCE-based objective, where we update embeddings by manually computed gradients.
> - (Reviewers 1 (koXd) & 2 (vWD3) & 3 (s1h6)) We add more descriptions in the main text according to reviewers.

---

### Author Response · Authors · 2022-11-28
**(General Response) Restatement of the Points and Comparison with Tensorized Embedding in NLP**

We just identify a NeurIPS spotlight paper and would like to re-state the following points.

**Techanical advantages:** By adopting tensorized embedding, we devise a new quantum-inspired embedding approach based on tensor decomposition. It achieves low memory cost yet the high model capacity for the problem of network embedding (see also the high compression ratio in our attached new table compared with tensorized embedding methods in NLP).

The high model capacity comes from the tensorized embeddings in high-dimensional Hilbert space instead of low-dimensional vectors adopted by conventional embedding methods (e.g. [3]), and the low memory cost is due to not only the direct use of tensor decomposition, but also (and more importantly) our inner-product-based objectives which can be simplified by our mathematical derivation (see Sec. 4.2 and Appendix C.1). It skips the step of calculating embeddings in Hilbert space and significantly reduces computation costs. While the counterpart works [1,2] are devoted to the NLP domain, and their compression ratio is much less than ours (see our below new table) because they only perform tensor factorization and still rely on the full tensorized embedding in the Hilbert space.

For a more in-depth comparison with [1,2], one may come up with a way for adapting our approach to the NLP in a way similar to converting the network embedding methods (e.g. [3]) to NLP method word2vec [4], where our reduction technique of full embedding as mentined above can still be applied. On the other hand, [1,2] can in general also be adapted to network embedding, e.g. by adopting GNN while the compression ratio can be much less than ours due to the above mentioned reason that they lack a mechanism to simplify the computation of full embeddings in Hilbert space besides decomposition.

**First attempt in the scalable network embedding domain:** We are the first to enable adopting the tensorized embedding for the network embedding domain, and further show its applicability and competitive performance on three most popular tasks: network reconstruction, node classification, and link prediction.

**Why it is quantum-inspired:** The reason why our approach is more close to quantum and has the potential to be run on quantum devices (yet [1,2] can not) is that i) we take into the "amplitude" (i.e. weights $w\_{ij}$ in Eq. 3) which makes the embeddings quantum states rather than tensors by CP decomposition in [1,2]. Differences between them are given as the below equations. ii) our method has no requirements for the deep neural network decoder, which is very hard to be implemented on quantum machines (and also to achieve quantum supremacy) due to the well-known barren plateaus in quantum neural networks [5].

-----

Quantum states in our paper (Eq. 3 in the paper): $\mathbf{x}\_{i} = \sum\_{b=1}^{B} w\_{ib}\bigotimes\_{c=1}^C \mathbf{u}\_{r(i,c)c}^b \text{s.t.} \|\mathbf{u}\_{r(i,c)c}^b\|=1$

Tensors by CP decomposition in [1,2]: $\mathbf{x}\_{i}=\sum\_{b=1}^{B}\bigotimes\_{c=1}^C \mathbf{u}\_{ic}^b$

In the table below, we compare our work with tensorized embedding methods in NLP. Note that no previous works in tensorized network embedding can be found.

|              | Compressive Ratio Compared to the Original Counterparts                                                                    | Outperforms Conventional SOTA Baselines | Area  | Tasks                                                               |
| ------------ | -------------------------------------------------------------------------------------------------------------------------- | -------------------------- | ----- | ------------------------------------------------------------------- |
| MorphTE [1]  | 81 times at most                                                                                                           | Nan                        | NLP   | machine translation, question answering, natural language inference |
| word2ket [2] | 93675 times at most                                                                                                        | Nan                        | NLP   | text summarization, language translation, question answering        |
| ours         | can be higher than $2^{32}/128=2^{25}\approx 3*10^7$ times (the original counterparts cannot be run on classical machines) | Yes                        | Graph | network reconstruction, link prediction, node classification        |

----

Ref:

[1] MorphTE: Injecting Morphology in Tensorized Embeddings. Gan et al. NeurIPS 2022 Spotlight.

[2] word2ket: space-efficient word embeddings inspired by quantum entanglement. Panahi et al. ICLR 2019 Spotlight.

[3] LINE: Large-scale Information Network Embedding. Tang et al. WWW 2015.

[4] Distributed representations of words and phrases and their compositionality. Mikolov et al. NIPS 2013.

[5] Barren plateaus in quantum neural network training landscapes. McClean et al. Nature Communications 2018.

---

### Decision · Program_Chairs · 2023-01-20

**Decision:**

Reject

**Justification For Why Not Higher Score:**

After authors' rebuttal, the reviewers still have major concerns about limited novelty and less convincing experiments. It seems to be combination of existing techniques, however its advantages are not clearly demonstrated by experiments.

**Justification For Why Not Lower Score:**

N/A

**Metareview: Summary, Strengths And Weaknesses:**

Summary: The paper proposes an unsupervised network (graph) embedding technique, which is a novel network embedding method inspired by quantum fields and modeled with tensors and contractions (i.e., outer product). In the experiment part, this paper showed superior performance on network analysis tasks and evaluated the parallelizability and interpretability.

Strengths: The model is interesting model and the combination with quantum is also very smooth and clearly discussed. This work is the first to enable adopting the tensorized embedding for the network embedding domain.

Weakness:  The contributions to network embedding are very limited. The embedding space is similar to word2ket to some extent, while the employed objective function is similar to LINE. Essentially, this paper is the combination of these two previous works.